

# New insights on the early Mesozoic evolution of multiple tectonic regimes in the northeastern North China Craton from the detrital zircon provenance of sedimentary strata

Yi Ni Wang[1], Wen Liang Xu[1,2], Feng Wang[1], Xiao Bo Li[1]

[1] College of Earth Sciences, Jilin University, Changchun, 130061, China
[2] Key Laboratory of Mineral Resources Evaluation in Northeast Asia, Ministry of Land and Resources of China, Changchun, 130061, China

*Correspondence to*: Wen Liang Xu (xuwl@jlu.edu.cn)

**Abstract.** To investigate the timing of deposition and provenance of early Mesozoic strata in the northeastern North China Craton (NCC), and to reconstruct the early Mesozoic tectono-paleogeography of the region, we combine LA–ICP–MS detrital zircon U–Pb dating, Hf isotopic data. Early Mesozoic strata include the Early Triassic Heisonggou, Late Triassic Changbai and Xiaoyingzi, and Early Jurassic Yihe formations. Detrital zircons in the Heisonggou Formation comprise ~58% Neoarchean to Paleoproterozoic and ~42% Phanerozoic grains that were sourced from areas to the south and north of the basins within the NCC. This indicates that Early Triassic deposition was controlled primarily by southward subduction of the Paleo-Asian oceanic plate beneath the NCC, and collision between the NCC and the Yangtze Craton (YC). Approximately 88% of sediments within the Late Triassic Xiaoyingzi Formation were sourced from the NCC to the south, with the remaining ~12% from the Xing'an–Mongol Orogenic Belt (XMOB) to the north. This implies that Late Triassic deposition was related to the final closure of the Paleo-Asian Ocean during the Middle Triassic and the rapid exhumation of the Su–Lu Orogenic Belt between the NCC and YC. In contrast, ~88% of sediments within the Early Jurassic Yihe Formation were sourced from the XMOB to the north, with the remaining ~12% from the NCC to the south. We therefore infer that rapid uplift of the XMOB and the onset of subduction of the Paleo-Pacific Plate beneath Eurasia occurred in the Early Jurassic.

## 1 Introduction

The tectonic evolution of the East Asian continental margin in the Mesozoic was controlled mainly by subduction of the Paleo-Pacific Plate beneath Eurasia. The evolution of the northeastern North China Craton (NCC), a component of the East Asian continental margin, was associated with the Paleo-Asian Ocean in the Paleozoic or early Mesozoic (Sengör and Natal'in, 1996; Li, 2006; Xu et al., 2013; Tang et al., 2013), overprinting of the Circum-Pacific Tectonic Domain in the Mesozoic (Lin et al., 1998; Li et al., 1999; Wu et al., 2000, 2004, 2007; Jia et al., 2004; Zhang et al., 2004; Shen et al., 2006; Xu et al., 2009, 2013), and subduction and collision between the NCC and the Yangtze Craton (YC) (Yang et al., 2007; Pei et al., 2008). Therefore, to determine the controls on early Mesozoic tectonics in the northeastern NCC, the timing of final





closure of the Paleo-Asian Ocean and the onset of subduction of the Paleo-Pacific Plate beneath Eurasia must be constrained. Final closure of the Paleo-Asian Ocean has been dated as late Permian (JBGMR, 1988; Shi, 2006), early–middle Permian to Middle Triassic (Wang et al., 2015b), and Early–Middle Triassic (Sun et al., 2004; Li et al., 2007; Cao et al., 2013). Similarly, multiple dates for the onset of subduction have been proposed, including early Permian (Ernst et al., 2007; Sun et

al., 2015; Yang et al., 2015; Bi et al., 2016), Late Triassic (Wu et al., 2011; Wilde and Zhou, 2015), latest Triassic to Early Jurassic (Zhou and Li, 2017), Early Jurassic (Xu et al., 2012, 2013; Tang et al., 2016; Guo, 2016; Wang et al., 2017).

These opinions were determined primarily through studies of early Mesozoic igneous rocks (especially granitoids) in NE China (Li et al., 1999; Wu et al., 2007; Zhao and Zhang, 2011; Xu et al., 2012, 2013). The diversity in the compositions of these granitoids resulted in the contrasting age determinations. Compared with studies of igneous rocks, few workers have

focused on sedimentary formation or paleogeographic reconstructions. Detrital zircon geochronology has recently become a powerful tool for provenance analysis, and can aid in constraining paleogeography, tectonic reconstructions, and crustal evolution (Cawood and Nemchin, 2001; Meng et al., 2010; Wang et al., 2012a).

The northeastern NCC contains a series of early Mesozoic faulted basins, referred to as the Fusong–Changbai basin group, which are filled by volcanics and coal-bearing clastic sediments. Compared with those in the northern and western–

central NCC (Meng, 2003; Meng et al., 2011, 2014; Li and Huang, 2013; Liu et al., 2015; Li et al., 2015; Xu et al., 2016; Meng, 2017), the evolution and dynamic setting of early Mesozoic basins in the northeastern NCC remain relatively unconstrained.

Modern geochronological data for Mesozoic basins in the northeastern NCC have not yet been acquired, and the establishment of a stratigraphic framework is based primarily on lithostratigraphic correlation. Due to the abundant

vegetative cover in the region and the lack of precise age data, questions remain regarding the formation of the Mesozoic deposits (JBGMR, 1963, 1976, 1988, 1997), including the following. 1) What was the timing of their formation? 2) What is the correct lithostratigraphic sequence of the units? 3) What is their relationship with the tectonic evolution of the region? In this contribution, we use U–Pb age and Hf isotopic data from detrital and magmatic zircons, combined with biostratigraphic data, to constrain the formation and provenance of early Mesozoic strata in the northeastern NCC, and reconstruct the

tectono-paleogeography of the region.

## 2 Geological background and sample descriptions

### 2.1 Geological background

The northeastern NCC is located at the intersection among three tectonic domains: the Paleo-Asian Tectonic Regime (also referred to as the Xing'an–Mongol Orogenic Belt (XMOB)) to the north, the Su–Lu Orogenic Belt to the south, and the

Circum-Pacific Tectonic Regime to the east (Fig. 1a). The Su–Lu Orogenic Belt and its eastward extension (the Jing–Ji belt in Korea) formed during early Mesozoic subduction and collision between the NCC and YC. The tectonic evolution of the



northeastern NCC is characterized by arc–continent collision in the early Paleozoic (Pei et al., 2014), late Paleozoic subduction of the Paleo-Asian Oceanic Plate beneath the NCC (Cao et al., 2012), and middle–late Permian to Middle Triassic closure of the Paleo-Asian Ocean (Sun et al., 2004; Li et al., 2009a; Wang et al., 2015b). Furthermore, the northeastern NCC was influenced by subduction, collision, and subsequent rapid exhumation between the NCC and YC in the early Mesozoic (Zheng et al., 2003; Yang et al., 2007; Pei et al., 2008; Liu et al., 2009). Subduction of the Paleo-Pacific Plate beneath Eurasia controlled the Mesozoic tectonic evolution of the East Asian continental margin (Xu et al., 2013). The Dunhua–Mishan and Yitong–Yilan faults occur in the northwestern part of the study area.

The northeastern NCC is composed primarily of Archean and Paleoproterozoic metamorphic basement (including the An'shan, Ji'an, and Laoling groups), Neoproterozoic and Paleozoic sedimentary cover sequences, and several Mesozoic basins (referred to as the Fusong–Changbai basin group) (Fig. 1a). Neoproterozoic strata comprise sandstone and limestone with minor stromatolites. Cambrian–Middle Ordovician strata are mainly epicontinental carbonate sediments, whereas late Carboniferous to early Permian units are characterized by marine and coal-bearing terrestrial sequences, which are unconformably overlain by Mesozoic volcano-sedimentary formation. Due to uplift of the craton in the middle Paleozoic, the region lacks Silurian–Devonian and early Carboniferous strata (SBGMR, 1989). Cenozoic basalts are common in the eastern part of the study area (JBGMR, 1988, 1997; Fig. 1b).

The interior of the NCC contains widespread Neoarchean to Paleoproterozoic magmatic rocks (Wu et al., 2007), but lacks evidence for Neoproterozoic or Paleozoic magmatism (except for early Paleozoic kimberlite) (JBGMR, 1988; LBGMR, 1989; IMBGMR, 1991). However, the northern margin of the NCC contains abundant Paleozoic igneous rocks (Zhang et al., 2004; Zhang et al., 2010; Cao et al., 2013; Pei et al., 2016). In the northeastern NCC, Paleozoic igneous rocks are concentrated to the north of the study area and are characterized by negative and positive $\varepsilon_{Hf}(t)$ values within the NCC and XMOB, respectively (Yang et al., 2006). The northern NCC contains Ordovician (467 Ma) medium-K calc-alkaline pyroxene andesite and middle Permian (270 Ma) garnet-bearing monzogranite with zircon $\varepsilon_{Hf}(t)$ values of –5.96 to –2.43 (Pei et al., 2016) and –17.1 to –14.1 (Cao et al., 2013), respectively. In contrast, the XMOB contains late Cambrian (493 Ma) low-K tholeiitic meta-diabase and late Permian (260 Ma) biotite monzogranite that yield zircon $\varepsilon_{Hf}(t)$ values of +9.42 to +14.89 (Pei et al., 2016) and +8.31 to +9.80 (Wang et al., 2015b), respectively.

Mesozoic magmatism was widespread along the East Asian continental margin (including the northeastern NCC and eastern XMOB). Within the northeastern NCC, Mesozoic magmatism occurred during the Late Triassic, Early Jurassic, Late Jurassic, Early Cretaceous, and Late Cretaceous (Yu et al., 2009; Xu et al., 2013; Wang et al., 2017). Mesozoic igneous rocks in the NCC and XMOB yield negative and positive zircon $\varepsilon_{Hf}(t)$ values, respectively (Yang et al., 2006; Pei et al., 2008; Wang et al., 2015b).

## 2.2 Early Mesozoic basin-filling sedimentary sequence

Mesozoic strata are well preserved and exposed in the study area, and are characterized by coal-bearing volcano-sedimentary formation containing abundant animal and plant fossils. The Fusong–Changbai Basin Group represents a series





of small- to medium-sized basins in the northeastern NCC. In this study, we focus on three small basins known as, from north to south, the Fusong, Yihe, and Yantonggou basins (Figs. 1b, 2). The Fusong Basin is located within Songshu Village, and is filled by the Late Triassic Xiaoyingzi Formation (herein, "Formation" is abbreviated to "Fm"), which is overlain by the Early Cretaceous Guosong Fm. The Yihe Basin, located in Yihe Village, is filled by the Late Triassic Changbai Fm and the Early Jurassic Yihe Fm. The Yantonggou Basin is located in Heisonggou Village and is filled by the Early Triassic Heisonggou Fm. These strata represent the early Mesozoic sedimentary sequence of the Fusong–Cangbai Basin group, and record deposition that was controlled by the tectonics of the northeastern NCC. The lithostratigraphic units are described in detail below (Fig. 2).

The Heisonggou Formation comprises conglomerate, sandstone, siltstone, and shale, and contains plant fossils. The stratotype is exposed in Heisonggou Village, strikes E–W, and extends westward into Korea (Fig. 1b, section 3). The stratotype displays faulted contacts with the under- and overlying Mesoproterozoic strata (JBGMR, 1963, 1976). The basal conglomerate contains clasts of quartzite, marble, phyllite, and schist derived from the underlying Mesoproterozoic units. Sandstone, siltstone, and shale are concentrated within the middle and upper strata. The upper part of the section is intruded by andesites (Fig. 3).

The Changbai Formation comprises a lower member of andesites and andesitic volcaniclastics, and an upper member of rhyolite and rhyolitic volcaniclastics. The stratotype is located in Naozhi Village. The formation overlies Paleoproterozoic strata with a faulted contact, and is unconformably overlain by the Early Jurassic Yihe Fm (JBGMR, 1997). The formation strikes E–W and is also observed in Naozhi and Yihe villages (Fig. 1b, section 2).

The Xiaoyingzi Formation comprises a ~30 m-thick lower member of conglomerate and sandstone (representing a depositional cycle) and an upper member of sandstone, siltstone, shale, mudstone, and coal. The conglomerate contains clasts of stromatolite-bearing dolomite that was sourced primarily from Neoproterozoic units. Allgovite is commonly observed to intrude along bedding planes. A thin layer of tuffaceous siltstone occurs in the middle of the section, and abundant plant and animal fossils are preserved within the middle and upper parts of the formation (Fig. 4). Volcanics and organic-rich beds have been used to constrain the timing of formation of the Xiaoyingzi Fm. The stratotype is exposed in Xiaoyingzi Village and the strata strike NW–SE (Fig. 1b, section 1). The base of the section is not exposed and the formation is overlain by Early Cretaceous volcanics (Guosong Fm) (JBGMR, 1963, 1976; Fig. 4).

The Yihe Formation comprises dominantly conglomerate, sandstone, siltstone, shale, coal, and minor tuffaceous siltstone (JBGMR, 1976, 1997, 1998; Fig. 5). The Yihe Fm unconformably overlies andesites of the Late Triassic Changbai Fm, which is present as gravels within the basal conglomerate of the Yihe Fm, thereby illustrating a conformable relationship between the two formations. The stratotype is exposed in Yihe Village and the strata strike E–W (Fig. 1b, section 2).

## 2.3 Sample descriptions

For U–Pb dating, we collected detrital and magmatic zircons from 10 early Mesozoic sandstone and igneous samples





(Fig. 2). Three samples were collected from the Heisonggou Fm, one from the Changbai Fm, three from the Xiaoyingzi Fm, two from the Yihe Fm, and one from the Guosong Fm, which overlies the Xiaoyingzi Fm. Details of their stratigraphy and petrography are presented in Figures 3–6 and are described below.

Sample 16LJ6-1 is a medium-grained feldspathic quartz sandstone from the lower Heisonggou Fm (Fig. 3). The sample is gray–white in color, displays detrital texture, and is bedded structure. Grains range in size from 0.3 to 0.6 mm and comprise plagioclase and alkali-feldspar (~14%), quartz (~78%), lithic fragments (~2%), and matrix (~5%) (Fig. 6a). Sample 15LJ4-11 is a fine-grained feldspathic quartz sandstone from the upper Heisonggou Fm. The sample is gray–white in color, exhibits detrital texture and is bedded structure. Grains are angular–subangular and range in size from 0.1 to 0.2 mm, comprising plagioclase and alkali-feldspar (~13%), quartz (~78%), lithic fragments (~2%), and calcareous cement (~6%) (Fig. 6b). Sample 15LJ4-6 is an andesite that intrudes the Heisonggou Fm (Fig. 3). It is light gray–green in color, displays pilotaxitic texture, and has massive structure (Fig. 6c).

Sample 15JFS1-1 is a medium-grained feldspathic quartz sandstone from the lower Xiaoyingzi Fm (Fig. 4). The sample is yellow–white in color, displays detrital texture, and is bedded structure. Grains range in size from 0.4 to 0.8 mm and comprise quartz (~80%),plagioclase and alkali-feldspar (~12%), lithic fragments (~3%, volcanic fragments), and matrix (~4%) (Fig. 6d). Sample 15JFS2-1 was collected from an allgovite dyke within the Xiaoyingzi Fm (Fig. 4). It is gray–green in color, displays porphyritic texture, and has massive structure. The phenocrysts are dominantly plagioclase (~5 vol.%) and the matrix displays pilotaxitic texture (Fig. 6e). Sample 16LJ8-1 is a tuffaceous siltstone from the middle Xiaoyingzi Fm (Fig. 4). The sample is white in color, displays detrital texture, and is bedded structure.

Sample 15JFS10-1 is a pyroxene andesite that unconformably overlies the Xiaoyingzi Fm. (Fig. 4). The andesite belongs to the Guosong Fm and is gray–green in color, displays porphyritic texture, and has massive structure (Fig. 6f). Sample 16LJ1-1 is an andesite from the Changbai Fm, which underlies the Yihe Fm. The andesite is gray–green in color, displays porphyritic texture, and has massive structure. The phenocrysts are mainly plagioclase (~10 vol.%). The matrix exhibits pilotaxitic texture (Fig. 6g).

Sample 15LJ1-2 is a fine-grained feldspathic quartz sandstone from the middle Yihe Fm (Fig. 5). It is gray–white in color, displays detrital texture, and is bedded structure. Grains comprise plagioclase and alkali-feldspar (~13%), quartz (~78%), lithic fragments (~3%), and matrix (~5%). Sample 16LJ3-1 is a tuffaceous siltstone from the upper Yihe Fm. It is white in color, displays detrital texture, and is bedded structure (Fig. 6h).

## 3 Analytical methods

### 3.1 Zircon U-Pb dating

Zircons were separated from samples using the conventional heavy liquid and magnetic techniques, and purified by handpicking under a binocular microscope at the Langfang Yantuo Geological Survey, Hebei Province, China. The handpicked zircons were examined under transmitted and reflected-light with an optical microscope, and in order to reveal their internal structures, cathodoluminescence (CL) images were obtained, using a JEOL scanning electron microscope



housed at the State Key Laboratory of Continental Dynamics, Northwest University, China. Distinct domains within the zircons were selected for analysis, based on their CL images. Agilent 7500a ICP-MS equipped with a 193 nm laser, housed at the State Key Laboratory of Geological Processes and Mineral Resources, China University of Geosciences (Wuhan), was used to measure the U-Pb age of zircons. Zircon 91500 was used as external standard for age calibration and the NIST SRM

610 silicate glass was applied for the instrument optimization. The crater diameter was 32 μm during the analyses. The instrument parameter and detail procedures were described by Yuan et al. (2004). The ICPMSDataCal 7.0 (Liu et al., 2010) and Isoplot 3.0 (Ludwig, 2003) programs were used for data reduction. Correction for common Pb was made following Andersen (2002). Errors on individual analyses by LA-ICP-MS are quated at the 1σ level, while errors on pooled ages are quoted at the 95 % (2 σ) confidence level. The dating results are presented in Table S1.

In addition, the samples of 15JFS2-1 (allgovite) and 15JFS10-1(pyroxene andesite), provide on 52 and 32 zircons grains, respectively. Considering the few zircon grains, measurement for the two samples were conducted using a Cameca 1280 SIMS at the Institute of Geology and Geophysics, Chinese Academy of Sciences in Beijing, using operating and data processing procedures similar to those described by Li et al. (2009b).

## 3.2 Hf isotopic analyses

*In situ* zircon Hf isotope analyses were conducted using a Neptune Plus MC-ICP-MS  (Thermo Fisher Scientific, Germany) equipped with a 193 nm excimer ArF laser ablation system (Lambda Physik, Göttingen, Germany) that was hosted at the State Key Laboratory of Geological Processes and Mineral Resources, China University of Geosciences. The energy density of laser ablation that was used in this study was 5.3 J cm$^{-2}$. Helium was used as the carrier gas within the ablation cell and was merged with argon (makeup gas) after the ablation cell. A simple Y junction downstream from the

sample was used to add small amounts of nitrogen (4 ml min$^{-1}$) to the argon makeup gas flow (Hu et al., 2008a, 2008b). Compared with the standard arrangement, the addition of nitrogen in combination with the use of a newly designed X skimmer and Jet sample cones in Neptune Plus, improved the signal intensities of Hf, Yb, and Lu by factors of 5.3, 4.0, and 2.4, respectively. All data were acquired using a single spot ablation mode with a 44 μm spot size. Each measurement consisted of 20 s of acquisition of the background signal followed by 50 s of ablation signal acquisition. Details of the

operating conditions for the laser ablation system and the MC-ICP-MS instrument and analytical method are given in Hu et al. (2012). The dating results are presented in Table S2.

## 4 Analytical results

### 4.1 Zircon U–Pb dating

#### 4.1.1 Heisonggou Formation

Zircons from sample 16LJ6-1 are euhedral–subhedral and display fine oscillatory growth zoning in cathodoluminescence (CL) images (Fig. 7a), indicating a magmatic origin. Ages from 68 analyses range from 248 to 2485





Ma (ages of >1000 Ma are $^{207}$Pb/$^{206}$Pb ages, whereas ages of <1000 Ma are $^{206}$Pb/$^{238}$U ages), yielding four major age populations that yield weighted mean ages of 252 ± 1 Ma (MSWD = 1, n = 21), 293 ± 2 Ma (MSWD = 1.2, n = 10), 323 ± 2 Ma (MSWD = 0.39, n = 15), and 2402 ± 12 Ma (MSWD = 8.5, n = 14) (Fig. 8a). Other grains yield ages of 339 (2 grains), 594, 1757, 1791, 2006, 2075, and 2171 Ma (Table S1). The youngest population age of 252 ± 1 Ma constrains the maximum
depositional age of this sample (i.e., the medium-grained feldspathic quartz sandstone was deposited after ~252 Ma).

Zircon grains from sample 15LJ4-11 are euhedral–subhedral and display fine oscillatory growth zoning in CL images (Fig. 7b), indicating a magmatic origin. Some grains are round and exhibit a core–rim texture. Ages from 72 analyses range from 248 to 2832 Ma and yield nine age populations with weighted mean ages of 253 ± 3 Ma (MSWD = 0.4, n = 7), 270 ± 3 Ma (MSWD = 0.24, n = 7), 304 ± 3 Ma (MSWD = 0.38, n = 5), 323 ± 3 Ma (MSWD = 0.12, n = 6), 360 ± 5 Ma (MSWD =
0.35, n = 3), 382 ± 7 Ma (MSWD = 0.03, n = 3), 1845 ± 9 Ma (MSWD = 1.7, n = 14), 2337 ± 23 Ma (MSWD = 4.8, n = 4), and 2504 ± 8 Ma (MSWD = 4.0, n = 20) (Fig. 8b). Other grains yielded ages of 426, 2152, and 2838 Ma (Table S1). The youngest population age of 253 ± 3 Ma represents the maximum depositional age of the fine-grained feldspathic quartz sandstone.

Zircons from sample 15LJ4-6 are euhedral–subhedral and display fine oscillatory growth zoning in CL images,
indicating a magmatic origin (Fig. 7c). Ages from 20 analyses range from 243 to 2435 Ma, yielding two populations with weighted mean ages of 246 ± 2 Ma (MSWD = 1.6, n = 8) and 297 ± 3 Ma (MSWD = 2.2, n = 6) (Fig. 8c). Other grains yielded ages of 268, 313, 1788, 1819, 2411, and 2435 Ma (Table S1). The mean age of 246 ± 2 Ma is interpreted as the crystallization age for the andesite, whereas the other ages are interpreted to captured zircons.

### 4.1.2 Changbai Formation

Zircon grains from sample 16LJ1-1 are euhedral–subhedral and display typical oscillatory growth zoning (Fig. 7d). Twenty-five analyses yielded a weighted mean $^{206}$Pb/$^{238}$U age of 227 ± 1 Ma (MSWD = 0.74, n = 25) (Fig. 8d; Table S1), which is interpreted as the crystallization age of the andesite.

### 4.1.3 Xiaoyingzi Formation

Zircons from sample 15JFS1-1 are euhedral–subhedral and show fine oscillatory growth zoning, indicating a magmatic
origin. Some rare grains are rounded and display fine oscillatory zoning (Fig. 7e). Ages from 79 analyses range from 220 to 3285 Ma, yielding five age populations with weighted mean ages of 224 ± 3 Ma (MSWD = 0.39, n = 7), 232 ± 5 Ma (MSWD = 0.27, n = 3), 257 ± 5 Ma (MSWD = 2.3, n = 3), 1880 ± 3 Ma (MSWD = 2.9, n = 42), and 1982 ± 21 Ma (MSWD = 1.2, n = 4) (Fig. 8e). Other grains yielded ages of 275, 277, 294, 296, 315, 364, 369, 392, 397, 433, 453, 496, 569, 691, 708, 1768, 2242, 2434, 2500, and 3285 Ma (Table S1). The youngest age population of 224 ± 3 Ma constrains the maximum
depositional age of the medium-grained feldspathic quartz sandstone.

Zircons from the allgovite (sample 15JFS2-1) are euhedral–subhedral and dominantly display oscillatory growth zoning, although some grains exhibit core–rim structures (Fig. 7f). Ages from 17 analyses range from 111 to 2516 Ma, with a peak





population at $113 \pm 2$ Ma (MSWD = 3.4, n = 5) (Fig. 8f). Other grains yield ages of 173, 218, 270, 332, 744, 1862, 1884, 1887, 1891, 2059, 2412, and 2516 Ma (Table S1). The youngest weighted mean $^{206}Pb/^{238}U$ age of $113 \pm 2$ Ma is considered to represent the age of intrusion of the allgovite.

Zircon grains from the tuffaceous siltstone (sample 16LJ8-1) are typically euhedral–subhedral and display oscillatory growth zoning in CL images, although some grains are rounded–subrounded (Fig. 7g). Ages from 65 analyses range from 223 to 2602 Ma, yielding six age populations with weighted mean ages of $227 \pm 2$ Ma (MSWD = 4.1, n = 3), $259 \pm 2$ Ma (MSWD = 4.5, n = 5), $1770 \pm 9$ Ma (MSWD = 1.9, n = 18), $1831 \pm 9$ Ma (MSWD = 2.1, n = 16), $2212 \pm 18$ Ma (MSWD = 7.7, n = 5), and $2486 \pm 15$ Ma (MSWD = 8.3, n = 7) (Fig. 8g). Other grains yielded ages of 240 (two grains), 316, 330, 505, 914, 1469, 1950, 2023, 2061, and 2602 Ma (Table S1). The youngest $^{206}Pb/^{238}U$ age of $223 \pm 2$ Ma is interpreted as the maximum depositional age of the tuffaceous siltstone.

### 4.1.4 Yihe Formation

Zircon grains from sample 15LJ1-2 are primarily euhedral–subhedral and display fine oscillatory zoning, whereas other grains are rounded and display oscillatory zoning (Fig. 7h), which, together with their Th/U ratios of 0.3–1.7, indicate a magmatic origin (Table S1). Ages from 77 analyses range from 177 to 2472 Ma, yielding six age populations with weighted mean ages of $184 \pm 2$ Ma (MSWD = 1.7, n = 7), $233 \pm 3$ Ma (MSWD = 0.82, n = 7), $245 \pm 1$ Ma (MSWD = 1.17, n = 17), $254 \pm 2$ Ma (MSWD = 0.72, n = 20), $266 \pm 2$ Ma (MSWD = 1, n = 11), and $1831 \pm 17$ Ma (MSWD = 9.9, n = 4) (Fig. 8h). Other grains yielded ages of 191, 200, 211, 286, 293, 323, 363, 460, 1468, 2471, and 2472 Ma (Table S1). The results indicate that the fine-grained feldspathic quartz sandstone was deposited after 184 Ma.

Zircon grains from sample 16LJ3-1 are typically euhedral–subhedral and display fine oscillatory growth zoning in CL images (Fig. 7i), indicating a magmatic origin. Ages from 65 analyses range from 178 to 2477 Ma, yielding three age populations with weighted mean ages of $182 \pm 1$ Ma (MSWD = 3.3, n = 20), $253 \pm 1$ Ma (MSWD = 0.74, n = 33), and $1831 \pm 20$ Ma (MSWD = 1.8, n = 4) (Fig. 8i). Other grains yielded ages of 212, 222, 237, 263, 276, 340, 2457, and 2477 Ma (Table S1). The youngest age population of $182 \pm 1$ Ma represents the maximum depositional age of the tuffaceous siltstone.

### 4.1.5 Guosong Formation

Zircon grains from the pyroxene andesite (15JFS10-1) are euhedral–subhedral and display oscillatory growth zoning and striped absorption in CL images, implying a magmatic origin (Fig. 7j). Ages from 14 analyses range from 112 to 1647 Ma, yielding two age populations with weighted mean ages of $113 \pm 3$ Ma (MSWD = 1.19, n = 2) and $227 \pm 3$ Ma (MSWD = 0.74, n = 7) (Fig. 8j). Other grains yielded ages of 156, 425, 438, 946, and 1647 Ma (Table S1). The youngest age of $113 \pm 3$ Ma is interpreted to represent the crystallization age of the pyroxene andesite.

### 4.2 Zircon Hf isotopes

We performed *in situ* Hf isotopic analysis on the same spots as used for U–Pb dating on samples from the Heisonggou

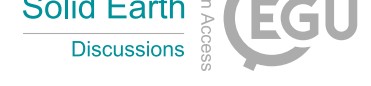

Fm (15LJ4-11), Xiaoyingzi Fm (15JFS1-1), and Yihe Fm (15LJ1-2). The results are listed in Table S2 and shown in Fig. 9.

### 4.2.1 Heisonggou Formation

We determined the Hf isotopic compositions of 22 detrital zircons from the Heisonggou Fm. Three zircon grains (~13.6%) with ages of 2513–2466 Ma yielded $\varepsilon_{Hf}(t)$ values of –4.2 to +4.9, four Paleoproterozoic (1862–1812 Ma) grains (~18.2%) yielded $\varepsilon_{Hf}(t)$ values of –1.9 to +0.2, and fifteen Phanerozoic (383–250 Ma) grains (~68.2%) yielded $\varepsilon_{Hf}(t)$ values of –16.3 to –7.0 (Table S2; Fig. 9a) and two-stage model ($T_{DM2}$) age of 2.3–1.7 Ga, similar to those of zircons from Phanerozoic igneous rocks in the NCC (Yang et al., 2006).

### 4.2.2 Xiaoyingzi Formation

We determined the Hf isotopic compositions of 25 detrital zircons with ages of 1928–221 Ma from the Xiaoyingzi Fm. Most zircon grains (~88%) yielded negative $\varepsilon_{Hf}(t)$ values ranging from –21.9 to –0.2 and $T_{DM2}$ ages of 2.7–1.6 Ga, whereas three zircon grains (~12%) with ages of 569, 496, and 453 Ma yielded positive $\varepsilon_{Hf}(t)$ values ranging from +2.9 to +8.5 and $T_{DM2}$ ages of 1.2–0.98 Ga (Table S2; Fig. 9b). Two grains within the latter with ages of 708 and 691 Ma yielded $\varepsilon_{Hf}(t)$ values of –10.9 and –6.1 and $T_{DM2}$ ages of 2.3 and 2.0 Ga, respectively.

### 4.2.3 Yihe Formation

We determined the Hf isotopic compositions of 17 detrital zircons with ages of 255 to 177 Ma from the Yihe Fm. Most grains (88%) yielded positive $\varepsilon_{Hf}(t)$ values of +2.4 to +12.6 and $T_{DM2}$ ages of 1.1 to 0.4 Ga, whereas two zircon grains (12%) with ages of 248 Ma yielded negative $\varepsilon_{Hf}(t)$ values of –5.8 and –1.9 and $T_{DM2}$ ages of 1.6 and 1.4 Ga (Table S2; Fig. 9c).

## 5 Discussion

### 5.1 Age of early Mesozoic strata in the northeastern NCC

Studies of Mesozoic strata in the northeastern NCC, and the establishment of their lithostratigraphic sequence, began during geological surveying in the 1960s–1970s. In the 1990s, these strata were reclassified and a general lithostratigraphic sequence was established by JBGMR (1997). Early Mesozoic strata in the study area include, from bottom to top, the Heisonggou Fm, Changbai Fm, Xiaoyingzi Fm, and Yihe Fm (Fig. 2). However, the timing and stratigraphic sequence of these Triassic–Early Jurassic units are based on lithostratigraphic correlations and remain controversial. Therefore, we have acquired new geochronological data for these units. We now combine the youngest concordant detrital zircon ages, the ages of interbedded volcanic and intrusive rocks, biostratigraphic ages, and the ages of overlying strata to constrain the age of the Mesozoic strata in the northeastern NCC.





### 5.1.1 Heisonggou Formation

The Heisougou Fm was first established in Heisonggou Village and was assigned to the Early Jurassic (JBGMR, 1963). It was subsequently reclassified as the Shiren Fm and assigned to the Early Cretaceous (JBGMR, 1997). Therefore, the age of the Heisonggou Fm remains uncertain. In this study, samples from the lower (16LJ6-1) and upper (15LJ4-11) Heisonggou

Fm yield youngest age populations of $252 \pm 1$ Ma and $253 \pm 3$ Ma, respectively, indicating that deposition of the Heisonggou Fm occurred after $252 \pm 1$ Ma. Furthermore, andesite intruding this unit (Fig. 3, sample 15LJ4-6) yielded a weighted mean age of $246 \pm 2$ Ma, thereby constraining the deposition of the Heisonggou Fm to between 252 and 246 Ma. This Early Triassic age contrasts with the previously proposed Early Jurassic (JBGMR, 1963) and Early Cretaceous ages (JBGMR, 1997).

### 5.1.2 Changbai Formation

The Changbai Fm was first established in Caiyuanzi and Ergulazi villages and was assigned to the Late Triassic. The formation comprises intermediate–acidic volcanics and has been subdivided into the Ergulazi Fm and Naozhigou Fm (JBGMR, 1976, 1997). Zircon U–Pb dating of andesite (16LJ1-1) from the Changbai Fm yielded a weighted mean age of $227 \pm 1$ Ma. Combined with a previously reported zircon $^{206}Pb/^{238}U$ age of $222 \pm 1$ Ma from the Naozhigou Fm in the

northern area of Caiyuanzi Village (Yu et al., 2009), we determine that the Changbai Fm was deposited in the Late Triassic.

### 5.1.3 Xiaoyingzi Formation

The Xiaoyingzi Fm was first established in Hengdaohezi and Xiaoyingzi villages and assigned to the Early Jurassic (JBGMR, 1971). It was later correlated with the Xiaohekou Fm and assigned to the Late Triassic (JBGMR, 1988, 1997). In contrast, JBGMR (2007) classified it within the Yihe Fm and assigned it to the Early Jurassic. Thus, the age of the

Xiaoyingzi Fm remains controversial. Detrital zircon grains from the medium-grained feldspathic quartz sandstone (15JFS1-1) and tuffaceous siltstone (16LJ8-1) yielded youngest concordant ages of $224 \pm 2$ Ma and $223 \pm 2$ Ma, respectively, suggesting that the Xiaoyingzi Fm was deposited after ~223 Ma. Freshwater bivalve fossils (e.g., *Ferganoconcha* sp. and *Sibireconcha* sp.) in this formation belong to the *Unio–Shaanxiconcha* assemblage (Zhu, 1991; JBGMR, 1997), similar to assemblages found in Late Triassic strata in China, South Australia, North American, and South Africa (Zhu, 1991). Plant

fossil assemblages (e.g., *Glossophyllum- Neocalamites*) in this formation are generally limited to the Late Triassic–Early Jurassic (JBGMR, 1997). We therefore conclude that the Xiaoyingzi Fm was deposited in the Late Triassic. The previously described "interbedded volcanic rocks" within the Xiaoyingzi Fm (JBGMR, 1971) are here reinterpreted as allgovite (15JFS2-1), which was intruded at $113 \pm 2$ Ma. Zircon grains from the pyroxene andesite (15JFS10-1) of the Guosong Fm also yield a formation age of $113 \pm 3$ Ma. These results indicate that the Guosong Fm, which overlies the Xiaoyingzi Fm,

and the allgovite have similar ages, and were produced during coeval magmatism.



### 5.1.4 Yihe Formation

The Yihe Formation was first established in Naozhigou and Yihe villages and was assigned to the Early Jurassic (JBGMR, 1976). Previous studies referred to this unit as the Yantonggou Fm and Shiren Fm (EGJS, 1975), and assigned it to the Late Jurassic and Early Cretaceous, respectively. Detrital zircon grains from the fine-grained feldspathic quartz sandstone (15LJ1-2) and tuffaceous siltstone (16LJ3-1) yielded youngest age populations of $184 \pm 2$ Ma and $182 \pm 1$ Ma, respectively. In addition, Early Jurassic plant fossils such as *Cladophlebis ukienesis*, *Marattia hoerensis*, and *Pterophyllum propinquum* are observed within the Yihe Fm (Si and Zhou, 1962). Thus, combined with the absence of Middle–Late Jurassic strata in southern Jilin Province, we conclude that the Yihe Fm was deposited in the Early Jurassic.

We have used our new geochronological data, field relationships, and biostratigraphic data to establish a new early Mesozoic stratigraphic framework for the northeastern NCC (Fig. 2), which is summarized as follows. The Heisonggou Fm is assigned to the Early Triassic (252–246 Ma) based on U–Pb ages of detrital and magmatic zircons. The Xiaoyingzi Fm is assigned to the Late Triassic based on geochronological data and fossil assemblages. Deposition of this unit was later than the Changbai Fm. The Yihe Fm was deposited in the Early Jurassic, consistent with biostratigraphic data (Si and Zhou, 1962).

### 5.2 Stratigraphic correlation with early Mesozoic strata in the northern NCC

Triassic–Middle Jurassic strata are well preserved and exposed in the northern NCC (Meng et al., 2013; Meng, 2017), and were deposited within early Mesozoic basins in the Yinshan–Yanshan orogenic belt (e.g., Beipiao, Xiabancheng, Jingxi, and Shiguaizi basins) (Fig. 10; Meng et al., 2018). The northern NCC shared a similar stratigraphic and sedimentological evolution to the interior of the craton in the Mesoproterozoic to Paleozoic, but has undergone a distinct evolution since the Mesozoic, involving alternating periods of contraction and extension (Davis et al., 2001; Cui et al., 2002).

Triassic–Middle Jurassic deposits in the northern NCC are dominated by volcano- sedimentary formations containing coal and abundant animal and plant fossils, and their stratigraphy, geochronology, and depositional processes are well constrained (Meng, 2003; Meng et al., 2011, 2014; Li and Huang, 2013; Liu et al., 2015; Li et al., 2015; Xu et al., 2016; Meng, 2017). Here, we determine the relationship between early Mesozoic strata in the northern and northeastern NCC. The late Permian Tiechang Fm in the northeastern NCC is characterized by gray–purple coarse sandstone, pebbly sandstone, siltstone, and shale, similar lithological association to those of late Permian strata of the northern NCC (e.g., the lower Tiechang Fm correlates with the Shihezi Fm, Naobaogou Fm, and Laowuopu Fm). In contrast, Early Triassic strata (e.g., the Heisonggou Fm) are only observed in the northeastern NCC. In the northern NCC, Middle Triassic strata are rare and only observed within the central area (e.g., the Xiabancheng Basin), and are characterized by sandstone, siltstone, and shale. Such strata are not observed in the northeastern NCC. Late Triassic strata (e.g., the Changbai Fm and Xiaoyingzi Fm) are commonly observed in the northeastern NCC, and comprise intermediate–acidic volcanics and coal-bearing clastic sediments. In the northern NCC, coeval and comparable strata are represented by the Xiaolanwuo Fm, Wuchang Fm, and Shanggu Fm



in the Xiabancheng Basin, as well as the Yangcaogou Fm in the Beipiao Basin. For example, the Late Triassic Xiaoyingzi Fm in the northeastern NCC contains similar lithological association and fossil assemblages to the Yangcaogou Fm and Shanggu Fm in the northern NCC. Furthermore, ages of volcanics in the Changbai Fm (227 and 222 Ma; Yu et al., 2009) are similar to those of volcanics in the Xiaolanwuo Fm (225 ± 1 Ma) and Wuchang Fm (227.6 ± 2 Ma) (Meng et al., 2018) in the

Xiabancheng Basin. The Early Jurassic Yihe Fm in the northeastern NCC is comparable to the lower Beipiao Fm, Xiahuayuan Fm, lower Yaopo Fm, and Zhaogou Fm in the northern NCC, as indicated by the presence of similar coal-bearing layers and plant fossil assemblages (JBGMR, 1988; LBGMR, 1989; HBGMR, 1989; BBGMR, 1991; IMBGMR, 1991).

     In summary, we suggest that similar late Paleozoic sedimentary formation (North China type) occur in both the

northern and northeastern NCC. However, Early Triassic strata are only observed within the northeastern NCC. In contrast, Middle Triassic strata are observed within the northern NCC but are absent in the northeastern NCC. Similar Late Triassic sedimentary formations and contemporaneous volcanics are observed in the northern and northeastern NCC. Earliest Jurassic strata are absent in the northern and northeastern NCC, whereas middle–late Early Jurassic strata are widespread in both regions, and are correlated through the observation of similar coal-bearing layers and plant fossils (JBGMR, 1988; LBGMR,

1989; HBGMR, 1989; BBGMR, 1991; IMBGMR, 1991). Furthermore, regional unconformities are identified between late Permian and Triassic strata, and Late Triassic and Jurassic strata in the northern and northeastern NCC (Fig. 10), respectively.

## 5.3 Provenance of early Mesozoic strata in the northeastern NCC

     A generally accepted method to identify the source of sedimentary units is to compare zircon U–Pb age and Hf isotopic

data with areas or units that may have supplied sediment to the region (Dickinson and Gehrels, 2008). Our 502 detrital zircon U–Pb analyses are grouped in three age populations, namely Neoarchean to Paleoproterozoic, Neoproterozoic, and Phanerozoic (Fig. 11). Magmatic zircon grains from igneous and sedimentary rocks within the northern margin of the NCC generally yield two age populations (Neoarchean to Paleoproterozoic and late Paleozoic) (Yang et al., 2006; Zhang et al., 2010), whereas igneous rocks and Paleozoic sediments from the southern margin of the XMOB contain mainly Phanerozoic

and minor Neoproterozoic zircon grains (Meng et al., 2010; Wang et al., 2012b; Wang et al., 2014). In addition, Phanerozoic zircons from the NCC typically yield negative $\varepsilon_{Hf}(t)$ values, whereas those from the southern margin of the XMOB typically yield positive $\varepsilon_{Hf}(t)$ values (Yang et al., 2006; Cao et al., 2013).

### 5.3.1 Early Triassic Heisonggou Formation

     Approximately 58% of detrital zircon grains from the Early Triassic Heisonggou Fm yield Neoarchean to

Paleoproterozoic ages (2.8–1.8 Ga), forming two peaks at ~2.5 and ~1.8 Ga, typical of the NCC (Ma and Wu, 1981; Zhao et al., 2001; Gao et al., 2004). In contrast, the ~42% magmatic zircon grains yield Phanerozoic ages (426–248 Ma). These Phanerozoic magmatic events are not recorded within the interior of the NCC (Yang et al., 2017), but are observed within



the northern margin of the NCC (Zhang et al., 2004; Zhang et al., 2010; Wu et al., 2011; Cao et al., 2013; Pei et al., 2014; Wang et al., 2015b; Wang et al., 2016). These observations, combined with negative $\varepsilon_{Hf}(t)$ values (–16.3 to –7.0) and two-stage model ($T_{DM2}$) age of 2.3–1.7 Ga, indicate that all sedimentary material within this formation was sourced from the NCC.

In addition, except for some rounded Neoarchean and Paleoproterozoic grains, detrital zircon grains (especially those with Phanerozoic ages) are euhedral–subhedral, suggesting a lack of long-distance transportation. This view is also supported by the short deposition time (between 252 and 246 Ma) of the Heisonggou Fm.

Based on the present distribution of the NCC basement and Phanerozoic igneous rocks in the northeastern NCC (JBGMR, 1988), we suggest that at least 42% of the sediment was sourced from Phanerozoic igneous rocks along the northern margin of the NCC, to the north of the basin. In contrast, Neoarchean and Paleoproterozoic sediment was likely sourced from regions surrounding the basin.

### 5.3.2 Late Triassic Xiaoyingzi Formation

Approximately 71% of detrital zircon grains from the Late Triassic Xiaoyingzi Fm yielded Neoarchean to Paleoproterozoic ages (with peaks at ~2.5 and ~1.88 Ga), indicating that they were sourced from the NCC (Gao et al., 2004). The other analyzed detrital zircon grains yielded ~26% Phanerozoic (496–220 Ma) and ~3% Neoproterozoic ages (914–569 Ma). Three zircon grains (age at 569, 496, and 453 Ma) yielded positive $\varepsilon_{Hf}(t)$ values (+2.9 to +8.5), whereas the other Phanerozoic and Neoproterozoic grains yielded negative $\varepsilon_{Hf}(t)$ values (–21.9 to –5.0), suggesting that the former were sourced from the XMOB and the latter from the NCC (Yang et al., 2006; Cao et al., 2013; Pei et al., 2014).

All Phanerozoic zircons are euhedral–subhedral, suggesting the Xiaoyingzi Fm sediments were not transported over long distances. Combined with the observation of Phanerozoic and minor Neoproterozoic igneous rocks to the north of the basin (JBGMR, 1988), we conclude that ~29% of sediments from the Xiaoyingzi Fm were sourced from an area to the north of the basin, with the remaining ~71% sourced from regions surrounding the basin.

### 5.3.3 Early Jurassic Yihe Formation

Approximately 91% of detrital zircon grains from the Early Jurassic Yihe Fm yielded Phanerozoic ages (460–177 Ma), with the remaining ~9% yielding Neoarchean to Paleoproterozoic ages (peaks at ~2.5 and ~1.8 Ga). The former are consistent with Phanerozoic magmatism along the northern margin of the NCC and XMOB, whereas the latter are typical of the NCC interior (Gao et al., 2004; Zhang et al., 2004; Yang et al., 2006; Cao et al., 2013; Wang et al., 2015b). The Hf isotopic analyses of 17 zircon grains with the Phanerozoic ages indicate that two grains (~12%) with ages of 248 Ma yielded $\varepsilon_{Hf}(t)$ values of –5.8 to –1.9, and the remaining 15 grains (~88%) with Phanerozoic ages yielded $\varepsilon_{Hf}(t)$ values of +2.4 to +12.6. We infer that the former, together with Neoarchean and Paleoproterozoic detrital zircon grains, were sourced from the NCC, whereas the latter were derived from the XMOB (Yang et al., 2006). Furthermore, all Phanerozoic zircon grains are





euhedral–subhedral, suggesting that the Phanerozoic sediments did not undergo long-distance transport. The age populations and $\varepsilon_{Hf}(t)$ values of the detrital zircon grains indicate that the Yihe Fm was sourced mainly from the XMOB to the north of the basin.

## 5.4 Reconstruction of the early Mesozoic tectono-paleogeography of the northeastern NCC

5        The early Mesozoic tectonic evolution of the northeastern NCC not only was influenced by subduction of the Paleo-Asian oceanic plate beneath the NCC, final closure of the Paleo-Asian Ocean to the north (Zhang et al., 2004; Wu et al., 2011; Cao et al., 2013), but also by subduction and collision between the NCC and YC to the south (Pei et al., 2008, 2011), as well as subduction of the Paleo-Pacific Plate beneath Eurasia to the east (Xu et al., 2013; Guo, 2016; Wang et al., 2017). However, the spatio-temporal extents of these influences, and the timing of final closure of the Paleo-Asian Ocean and the
onset of subduction of the Paleo-Pacific Plate remain controversial. Here we use provenance and changes in provenance within early Mesozoic strata of the northeastern NCC to reconstruct the early Mesozoic tectono-paleogeography of the northeastern NCC.

### 5.4.1 Early Triassic: southward subduction of the Paleo-Asian oceanic plate, and subduction and collision between the NCC and YC

Approximately 42% of the Early Triassic Heisonggou Fm sediments were sourced from the northern margin of the NCC; no evidence is seen for XMOB-sourced grains. We therefore infer that in the Early Triassic, southward subduction of the Paleo-Asian oceanic plate beneath the NCC was ongoing (i.e., final closure of the Paleo-Asian Ocean had not yet occurred) (Cao et al., 2013; Wang et al., 2015b). Subduction resulted in uplift along the northern margin of the NCC, producing a paleogeographic highland that acted as a source during deposition of the Heisonggou Fm (Fig. 12a).
The remaining ~58% of sediments of the Heisonggou Fm were sourced from Neoarchean and Paleoproterozoic NCC basement, which is observed in areas surrounding the basin, as well as to the south. We infer that the area to the south of the basin was uplifted in the Early Triassic, consistent with subduction and collision between the NCC and YC at this time (Pei et al., 2008, 2011; Liu et al., 2012; Zheng et al., 2013).

Thus, we conclude that southward subduction of the Paleo-Asian oceanic plate and subduction and collision between
the NCC and YC resulted in uplift of the northern and southeastern margins of the NCC, respectively. These uplifted highlands provided sources for the Heisonggou Fm (Figs. 10, 12a).

### 5.4.2 Late Triassic: final closure of the Paleo-Asian Ocean and post-collisional exhumation of the Su−Lu Orogenic Belt

Zircon Hf isotopic compositions indicate that ~29% of sediments of the Late Triassic Xiaoyingzi Fm were sourced from
areas to the north of the basin (~4% from the XMOB and ~25% from the northern margin of the NCC). The presence of the XMOB material indicates that the Paleo-Asian Ocean had closed by the Late Triassic. We therefore suggest that final closure



of the ocean occurred in the Middle Triassic, which is also supported by the occurrence of Middle Triassic syn-collisional granitoids in the Yanbian region (Wang et al., 2015b), as well as the absence of Middle Triassic sedimentary strata in the northeast NCC (JBGMR, 1988).

Approximately 71% of Xiaoyingzi Fm zircon grains yield Neoarchean and Paleoproterozoic ages, suggesting they were
sourced from NCC basement. As NCC basement dominantly crops out to the south of the basin, we conclude that the region to the south of the basin, which still was a highland at the time, provided the primary source during deposition of the Xiaoyingzi Fm. Furthermore, we suggest that Late Triassic rapid exhumation of ultrahigh-pressure metamorphic rocks within the Su–Lu Orogenic Belt (Zhao et al., 2001; Zheng et al., 2013) produced the inferred uplift in the region to the south of the basin (Figs. 10, 12b).

**5.4.3 Early Jurassic: rapid uplift of the XMOB and the onset of subduction of the Paleo-Pacific Plate beneath Eurasia**

The provenance of the Early Jurassic Yihe Fm changed rapidly from ~71% (the Late Triassic Xiaoyingzi Fm) Neoarchean and Paleoproterozoic basement from an area to the south of the basin in the Late Triassic, to ~91% Phanerozoic rocks from an area to the north of the basin in the Early Jurassic. In addition, Hf isotopic compositions of Phanerozoic detrital zircon grains indicate that >88% of the grains were sourced from the XMOB, suggesting that rapid uplift of the
XMOB occurred during the Early Jurassic. The XMOB then became the dominant source during Early Jurassic deposition (Figs. 10, 12c).

Furthermore, Early Jurassic detrital zircons with the magmatic origin are common within the Yihe Fm, suggesting that Early Jurassic igneous rocks represented a significant source for the Yihe Fm. Early Jurassic igneous rocks are observed to the northeast of the basin within the northeastern NCC (Wu et al., 2011; Yu et al., 2012; Xu et al., 2013; Guo, 2016; Wang et
al., 2017), indicating that the northeastern area to the basin formed a highland in the Early Jurassic.

The above results, combined with the existence of Late Triassic extensional environment in eastern Heilongjiang Province (Xu et al., 2009; Wang et al., 2015a), indicate that Early Jurassic uplift in the northeastern NCC was related to oblique subduction of the Paleo-Pacific Plate beneath Eurasia. This interpretation is also supported by the presence of an Early Jurassic accretionary complex (Li et al., 1999; Wu et al., 2007; Zhou et al., 2009) and calc-alkaline igneous rocks in
eastern Heilongjiang and Jilin provinces (Yu et al., 2012; Xu et al., 2013; Guo, 2016; Wang et al., 2017).

In summary, Early Triassic deposition in the northeastern NCC was controlled by southward subduction of the Paleo-Asian oceanic plate, as well as subduction and collision between the NCC and YC (Fig. 12a). The absence of Middle Triassic strata in the northeastern NCC suggests final closure of the Paleo-Asian Ocean at this time, which is also supported by the provenance of the Late Triassic deposits. The presence of XMOB material in Late Triassic deposits suggests that the
Paleo-Asian Ocean had closed by this time. Rapid exhumation of the Su–Lu Orogenic Belt may have resulted in the formation of a paleogeographic highland within the southeastern margin of the NCC (Fig. 12b). A sudden change in provenance occurred in the Early Jurassic, which was likely related to rapid uplift of the XMOB and the onset of subduction of the Paleo-Pacific Plate beneath Eurasia (Fig. 12c).





**6 Conclusions**

Based on the U–Pb ages of detrital and magmatic zircons, detrital zircon Hf isotopic data, and biostratigraphic records from early Mesozoic strata of the northeastern NCC, we draw the following conclusions.

(1) The early Mesozoic stratigraphic sequence of the northeastern NCC comprises, from bottom to top, the Early Triassic (252–246 Ma) Heisonggou Formation, Late Triassic Changbai Formation (~227 Ma), Late Triassic Xiaoyingzi Formation, and Early Jurassic Yihe Formation.

(2) The provenance of Early Triassic Heisonggou Formation was all from within the NCC, with ~42% of sediment from an area to the north of the basin and ~58% from the area surrounding the basin. Approximately 88% of sediments of the Late Triassic Xiaoyingzi Fm were sourced from the NCC, whereas the remaining ~12% were derived from the XMOB. In contrast, >88% of sediments of the Early Jurassic Yihe Fm were sourced from the XMOB, with only ~12% from the NCC.

(3) Early Triassic deposition was controlled by both southward subduction of the Paleo-Asian oceanic plate beneath the NCC and northward subduction and collision between the NCC and YC. The Late Triassic deposition could be related to final closure of the Paleo-Asian Ocean and rapid exhumation of the Su–Lu Orogenic Belt between the NCC and YC. The sudden change in provenance recorded by Early Jurassic sediments, as well as the observations of coeval calc-alkaline volcanism and an accretionary complex, implies the rapid uplift of the XMOB and the onset of subduction of the Paleo-Pacific Plate beneath Eurasia.

(4) Final closure of the Paleo-Asian Ocean likely occurred in the Middle Triassic, consistent with the lack of Middle Triassic strata in the northeastern NCC and the observations of Middle Triassic syn-collisional granitoids along the Changchun–Yanji suture belt.

*Data availability*. Original data underlying the material presented are available by contacting the authors.

*Supplements*. Supplement information; Table S1; Table S2.

*Author contributions*. Yini Wang, designed the format and wrote the main content. Wenliang Xu designed the entire project and produced the title. Feng Wang participated in the field work and data analysis. Xiaobo Li modified and proofed the figures and captions.

*Competing interests*. The authors declare that they have no conflict of interest.

*Acknowledgements*. We appreciate the editor and anonymous reviewers for their constructive and valuable comments. We would like to thank the staff of the State Key Laboratory of Geological Processes and Mineral Resources, China University of Geosciences, Wuhan, China, for helping with LA-ICP-MS zircon U-Pb dating and zircon Hf isotope analyses. Meanwhile, we thank Prof. Fu-Hong Gao for sedimentary rocks identification. This study was supported by the National Natural Science Foundation of China (Grants 41330206 and 41702030), and supporting data are included in supporting information.

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

**Figure captions**

Figure 1: (a) Geological sketch map of the northeastern North China Craton. SC: Siberian Craton; CAOB: Central Asian Orogenic Belt; SCC: South China Craton. (b) Geological map of the Linjiang area showing the early Mesozoic strata and the three
study sections.

Figure 2: Early Mesozoic stratigraphy of the northeastern NCC, sampling sites, and dating results.

Figure 3: Stratotype outcrop of the Heisonggou Formation and sampling locations. The location of the stratotype is indicated by the dotted line in Fig. 1b (section 3).

Figure 4: Stratotype outcrop of the Xiaoyingzi Formation and sampling locations. The location of the stratotype is indicated by the
dotted line in Fig. 1b (section 1).

Figure 5: Stratotype outcrop of the Yihe Formation and sampling locations. The location of the stratotype is indicated by the dotted line in Fig. 1b (section 2).

Figure 6: Photomicrographs of selected early Mesozoic samples (cross-polarized light). (a) Sample 16LJ6-1, a medium-grained feldspathic quartz sandstone from the Heisonggou Fm; (b) Sample 15LJ4-11, a fine-grained feldspathic quartz sandstone from the
Heisonggou Fm; (c) Sample 15LJ4-6, an andesite that intrudes the Heisonggou Fm; (d) Sample 15JFS1-1, a medium-grained feldspathic quartz sandstone from the Xiaoyingzi Fm; (e) Sample 15JFS2-1, an allgovite that intrudes the Xiaoyingzi Fm; (f) Sample 15JFS10-1, a pyroxene andesite from the Guosong Fm; (g) Sample 16LJ1-1, an andesite from the Changbai Fm; (h) Sample 16LJ3-1, a tuffaceous siltstone from the Yihe Fm. Af: alkali-feldspar; Pl: plagioclase; Px: pyroxene; Q: quartz.





**Figure 7: Cathodoluminescence (CL) images of selected zircon grains from early Mesozoic strata. White circles indicate the locations of U–Pb dating analyses and blue circles show the locations of in situ Hf analyses. Values under and upper the images indicate correspinding zircon U-Pb age and measured εHf(t) values, respectively.**

**Figure 8: U–Pb concordia diagram for zircon grains from sedimentary and igneous rocks within early Mesozoic strata.**

**Figure 9: Hf isotopic compositions of detrital zircon grains from early Mesozoic strata of the northeastern NCC. XMOB: Xing'an– Mongol Orogenic Belt; YFTB: Yanshan Fold-and- Thrust Belt (Yang et al., 2006).**

**Figure 10: Correlation between the early Mesozoic stratigraphy of the northern and northeastern NCC (the stratigraphic sequence of the northern NCC is from Meng et al., 2018).**

**Figure 11: Relative probability diagram of detrital zircon grains from early Mesozoic strata of the northeastern NCC.**

**Figure 12: Tectono-paleogeography of northeastern China in the early Mesozoic, showing the paleogeographic evolution and provenance of early Mesozoic basins. (a) Early Triassic; (b) Late Triassic; (c) Early Jurassic. JJOB: Jing-Ji Orogenic Belt; JM: Jiamusi Massif; KM: Khanka Massif; MOO: Mongol-Okhotsk Ocean; NCC: North China Craton; PAO: Paleo-Asian Ocean; PPO: Paleo-Pacific Ocean; SS: South Siberia; SZM: Songne-Zhangguangcai Range Massif; XMOB: Xing'an-Mongol Orogenic Belt; YC: Yangtze Craton.**



# Figure 1

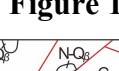



**Figure 2**

| Fusong-Changbai Basin Group | | | The early Mesozoic strata in the northeastern NCC | | | | |
|---|---|---|---|---|---|---|---|
| Time | Yantonggou Basin | Yihe Basin | Fusong Basin | Formation | Stratigraphic column | Sample | Lithologic characteristic | Fossil |
| K₁ | | | Guosong Fm | **Guosong Fm** | | ←15JFS10-1 (pyroxene andesite;113±3 Ma;SIMS) | | |
| J₂₋₃ | | | | | | *Section 1* | | |
| J₁ | | Yihe Fm | | **Yihe Fm** | | ←16LJ3-1 (tuffaceous siltstone;182±1 Ma;LA-ICP-MS) ←15LJ1-2 (fine-grained feldspathic quartz sandstone; 184±2 Ma; LA-ICP-MS) *Hf isotope* *Section 2* | Conglomerate, sandstone, siltstone, shale, coal beds, and interlays with tuffaceous siltstone. | Plant *Coniopters-Phoenicopsis* (Si & Zhou, 1962) |
| T₃ | | Xiaoyingzi Fm | | **Xiaoyingzi Fm** | | ←16LJ8-1 (tuffaceous siltstone; 223±2 Ma; LA-ICP-MS) ←15JFS2-1 (allgovite; 113±2 Ma; SIMS) ←15JFS1-1 (medium-grained feldspathic quartz sandstone; 224±3 Ma; LA-ICP-MS) *Hf isotope* *Section 1* | The upper member: sandstone, siltstone, shale, mudstone, and coal beds; The lower member: conglomerate and sandstone; Allgovites. | Bivalve *Unio-Shaanxiconcha* (Zhu,1991; JBGMR,1997) Plant *Glossophyllum-Neocalamites* |
| | | Changbai Fm | | **Changbai Fm** | | ←222±1 Ma (rhyolite;Yu et al., 2009) ←16LJ1-1 (andesite; 227±1 Ma;LA-ICP-MS) *Section 2* | The upper member: rhyolite and rhyolitic volcaniclastic rock; The lower member: andesite and andesitic volcaniclastic rock. | |
| T₂ | | | | | | | | |
| T₁ | Heisonggou Fm | | | **Heisonggou Fm** | | ←15LJ4-6 (andesite; 246±2 Ma; LA-ICP-MS) ←15LJ4-11(fine-grained feldspathic quartz sandstone; 253±3 Ma; LA-ICP-MS) *Hf isotope* ←16LJ6-1 (medium-grained feldspathic quartz sandstone; 252±1 Ma; LA-ICP-MS) *Section 3* | Conglomerate, sandstone, siltstone, shale, and contains plant fossils. | |

Legend:

- silt shale
- siltstone
- sandstone
- feldspathic quartz sandstone
- feldspathic sandstone
- glutenite
- conglomerate
- coal
- tuffaceous siltstone
- andesitic breccia
- rhyolitic and tuffaceous breccia
- andesite
- allgovite
- pyroxene andesite
- rhyolite
- plant fossil
- bivalve fossil



## Figure 3

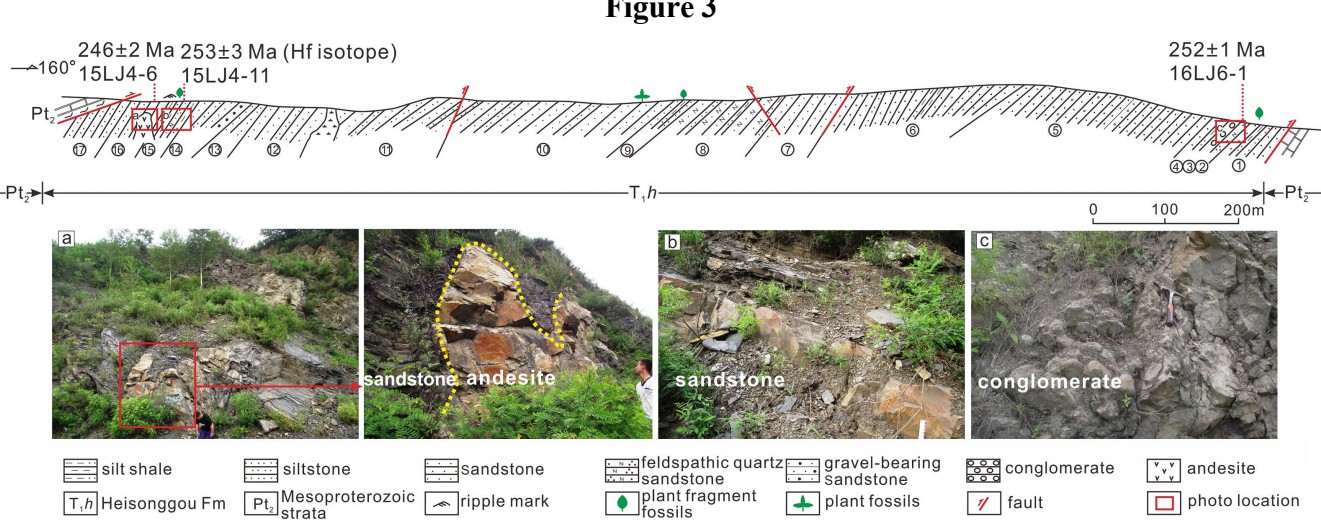

silt shale | siltstone | sandstone | feldspathic quartz sandstone | gravel-bearing sandstone | conglomerate | andesite

$T_1h$ Heisonggou Fm | $Pt_2$ Mesoproterozoic strata | ripple mark | plant fragment fossils | plant fossils | fault | photo location



**Figure 4**







# Figure 5

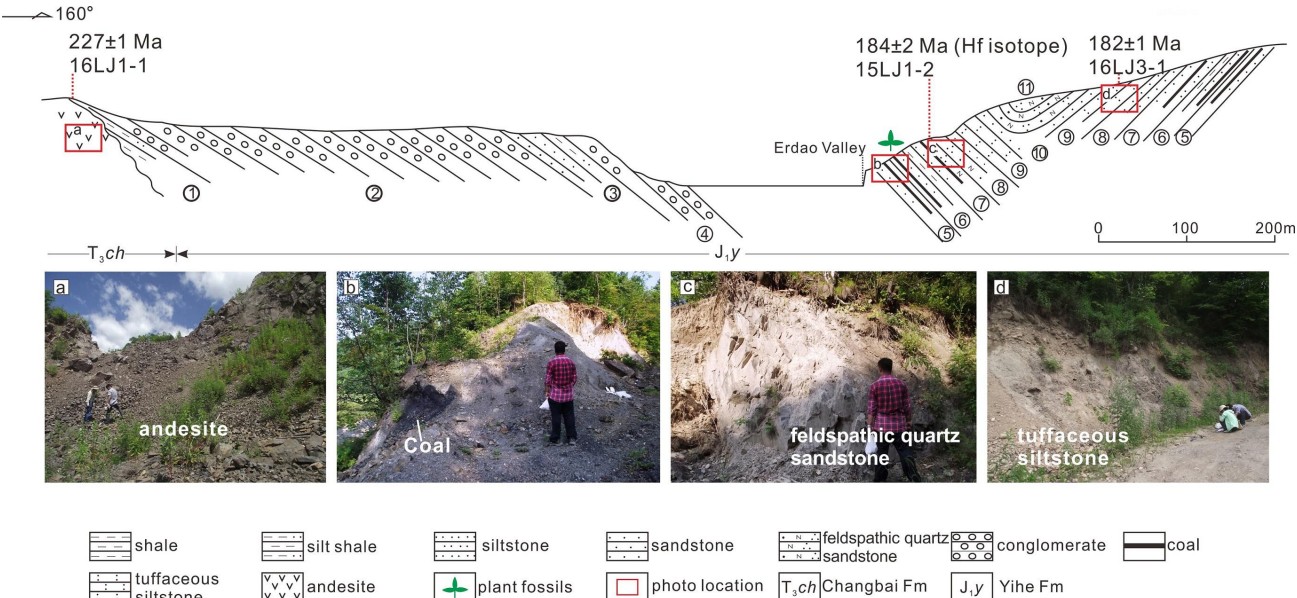





**Figure 6**

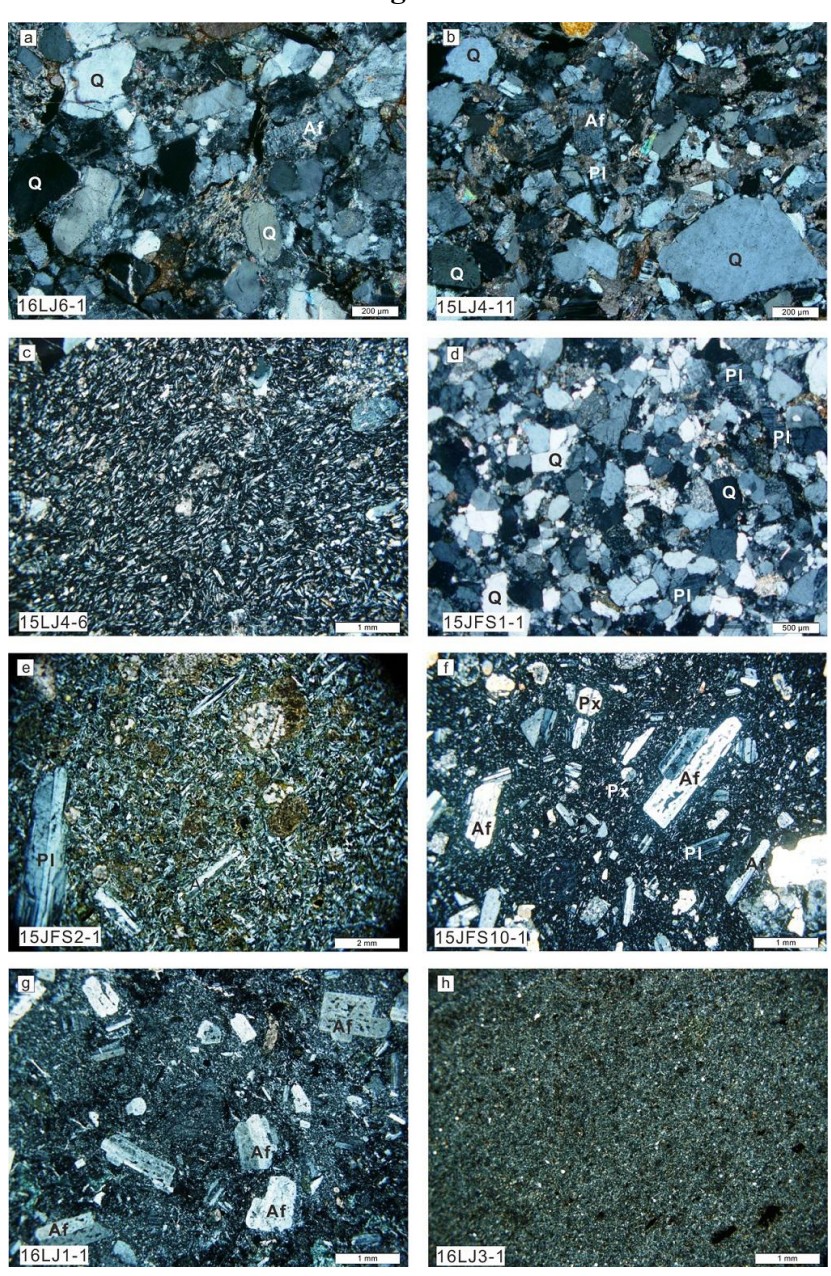



# Figure 7

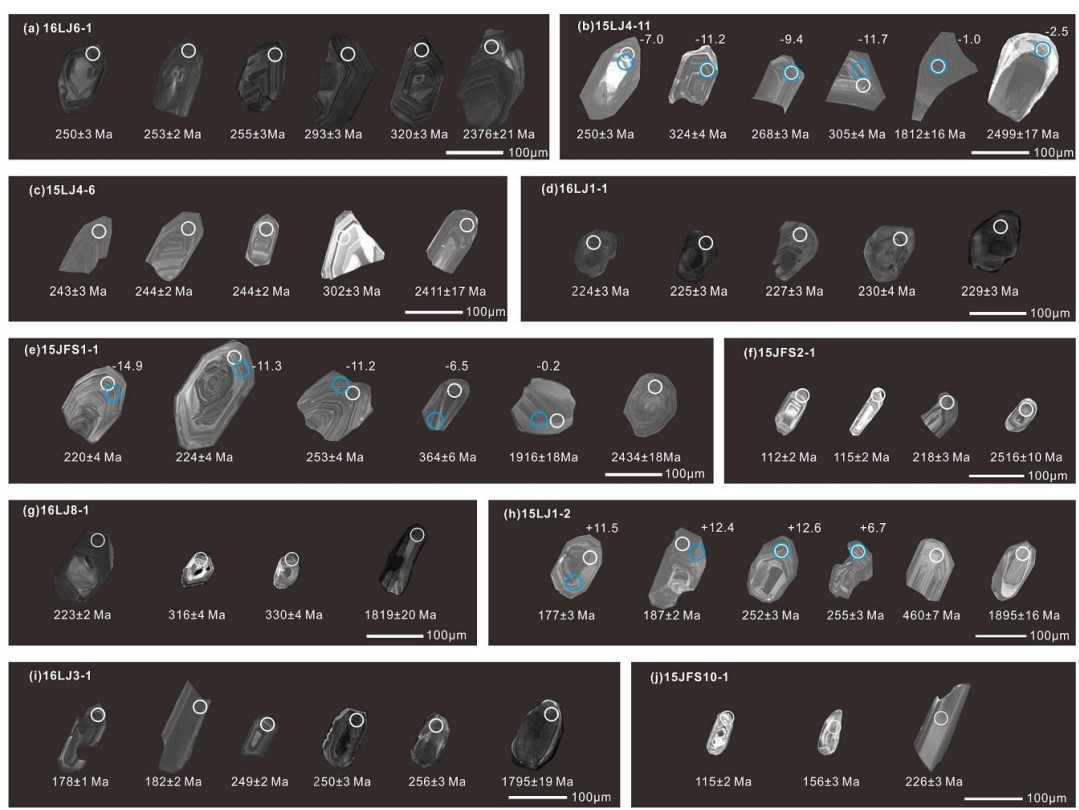



# Figure 8



**Figure9**

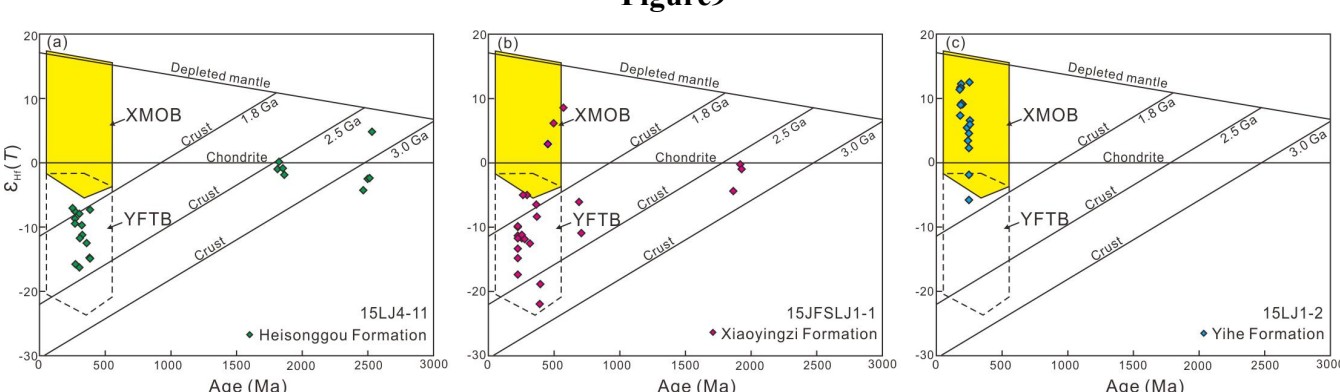





**Figure 10**

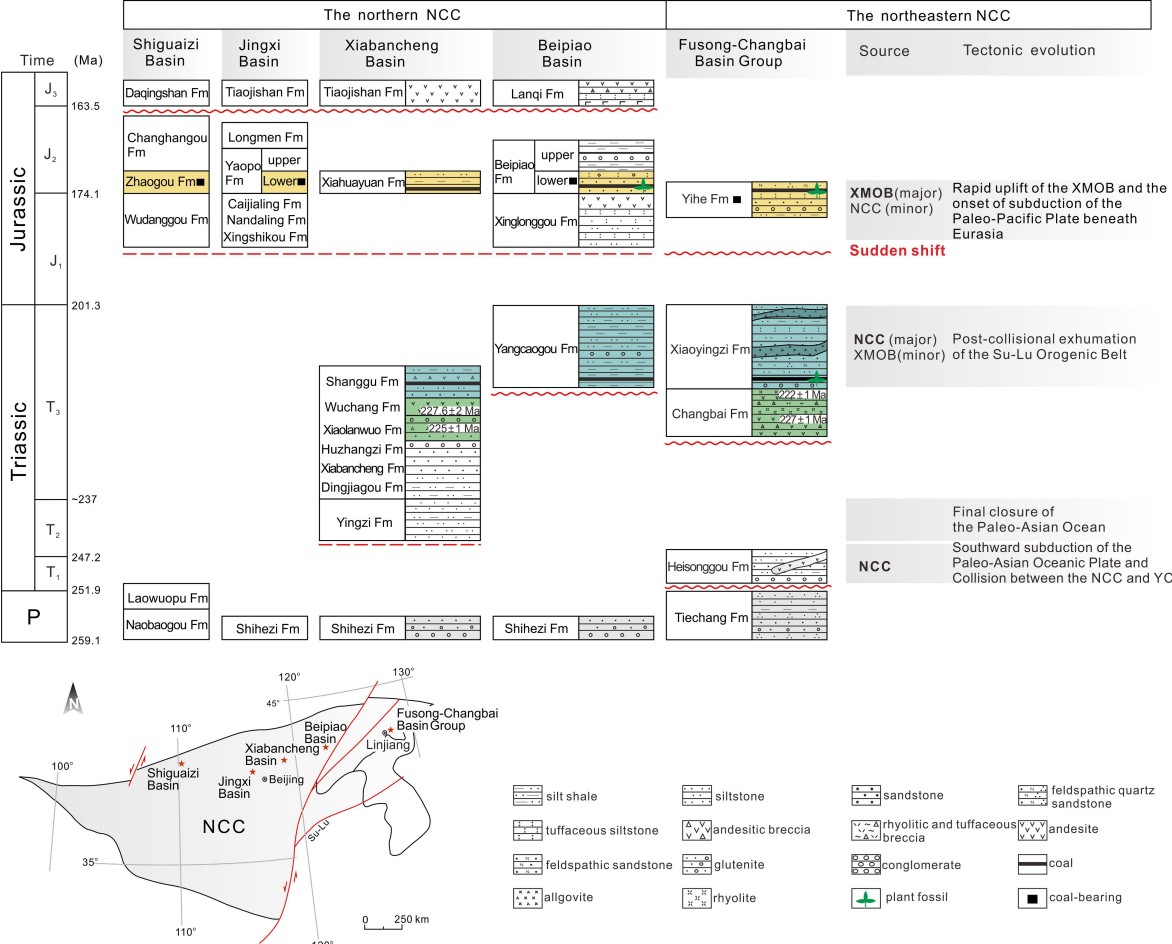



## Figure 11

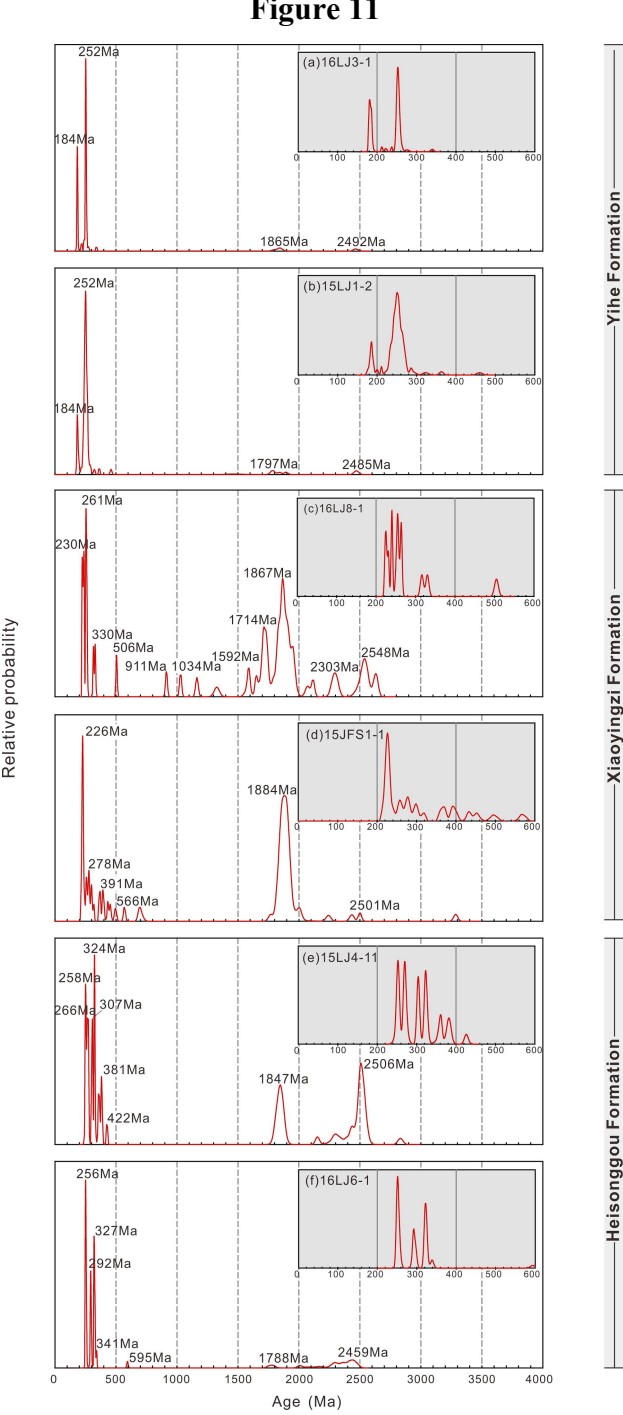




# Figure 12

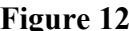

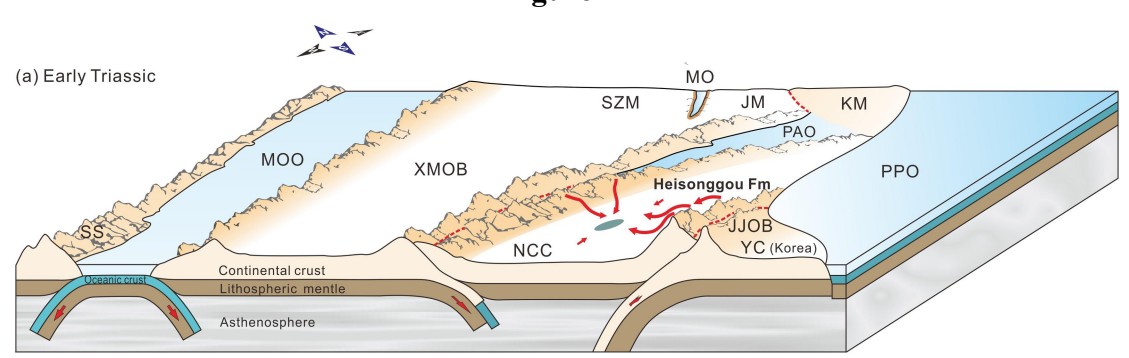

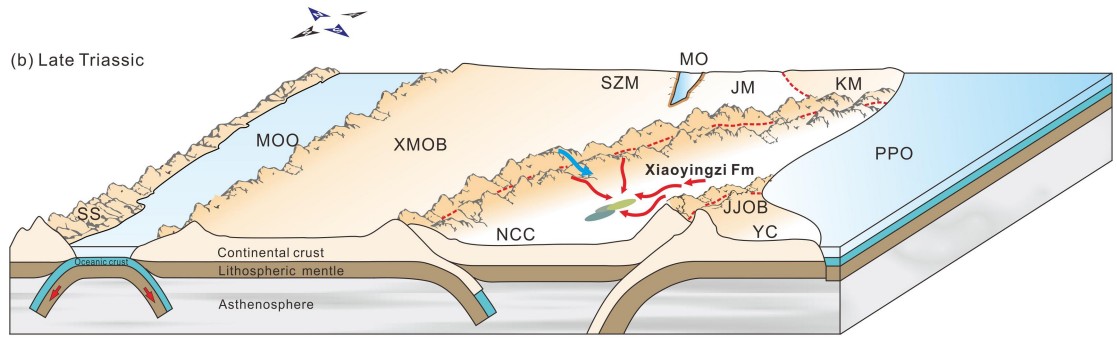

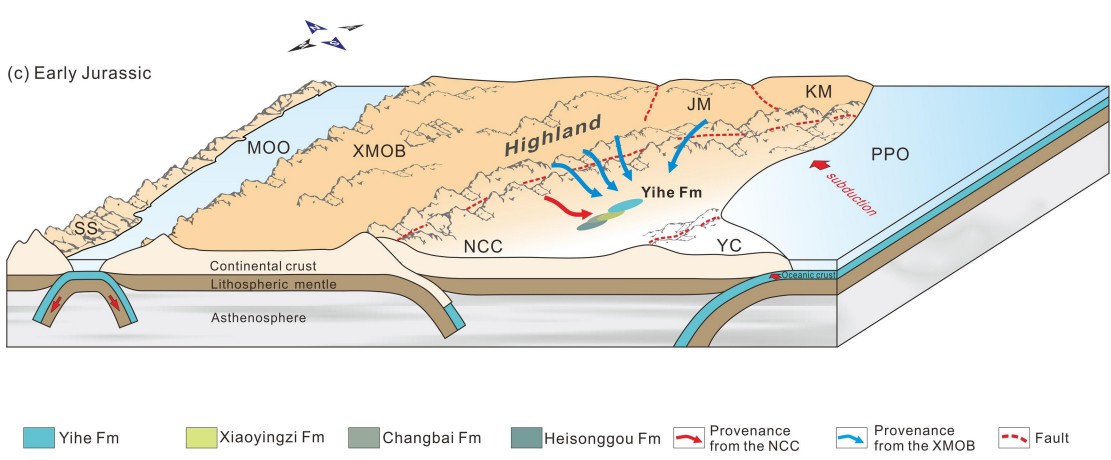