# Peer review of "New insights on the early Mesozoic evolution of multiple tectonic regimes in the northeastern North China Craton from the detrital zircon provenance of sedimentary strata"

_Solid Earth, 2018_

## Referee Comment (RC1) · Anonymous Referee #1 · 31 Aug 2018

The timing for final closure of the Paleo-Asian Ocean and onset of subduction of the Paleo-Pacific Plate beneath Eurasia is a very important topic in eastern Asian geological researches, but is still poorly constrained. In this manuscript, the authors presented new detrital zircon U-Pb dating and Hf isotopic data of the Early Mesozoic strata in northeastern North China Craton, with an aim to constrain their deposition ages and provenances. The authors proposed that onset of subduction of the Paleo-Pacific Plate beneath Eurasia occurred during the Early Jurassic and the final closure of the Paleo-Asian Ocean likely occurred in the Middle Triassic. The zircon U-Pb ages and Hf

isotopic data are in good quality. Therefore, the manuscript is worthy of publication in Solid Earth Discussions after a moderate revision. My comments and suggestions are listed below: 1. In several places (Page 1, Line 10; Page 2, Lines 24-25), the authors mentioned their using detrital zircon data "to reconstruct the early Mesozoic tectono-paleogeography of the region". This is a very difficult task. Although the detrital zircon U-Pb data presented in the manuscript is good in quality and valuable, there is no any field data from sedimentary analyses (such as paleocurrent direction, sedimentary facies). Therefore, I don't think that the authors can reconstruct the early Mesozoic tectono-paleogeography only by their limited detrital zircon data from the early Mesozoic strata. 2. In the Abstract (Page 1, Lines 11-13, 16-17, 19-20), Conclusions (Page 16, 7-10) and sections "5.3 Provenance of early Mesozoic strata in the northeastern NCC" and 5.4 Reconstruction of the early Mesozoic tectono-paleogeography of the northeastern NCC", the authors tried to use the percentages of detrital zircon U-Pb ages to give a quantitative analysis of sedimentary provenances. However, since zircon is only a very minor mineral in the sedimentary rocks, 42% detrital zircons with characteristics of the northern margin of the NCC do not mean that the 42% sediments were sourced from the northern margin of the NCC. Therefore, I don't think you can give a quantitative constraints on the sedimentary provenances of early Mesozoic sedimentary rocks. 3. Page 2, Lines 16-17: Words "poorly constrained" are better than "relatively unconstrained". 4. Page 3, Line 17: Neoproterozoic mafic magmatism around 900 Ma have been identified from the NCC in recent years by many studies (Liu et al., 2006, Chin. Sci. Bull. 51, 2375–2382; Gao et al., 2010, Geol. Bull. China 29, 1113–1122; Peng et al., (011, Lithos 127, 210–221; 2011, Gondwana Res. 20, 243–254; Wang et al., 2012, Sci. China Earth Sci. 55, 1461–1479; Zhang et al., 2016, Precambrian Research 272, 203–225). 5. Page 5, Lines 5-16: Is this the volume percentage of minerals? If then, please change "%" to "vol. %". 6. Page 16-25: In the reference list, the authors list almost 22 papers from their group. I suggest the authors add some relevant references from other groups. 7. Figure 10: Check the strata column of Xiabancheng Basin. Xingshikou and Nandaling formations are common in the

Xiabancheng Basin, but is not shown in the strata column.

---

## Referee Comment (RC2) · W. Xiao (Referee) · 12 Sep 2018

General comments The early Mesozoic evolution of multiple tectonic regimes in the northeastern North China Craton is of key importance for a better understanding of the tectonics of E Asia including the final closure of the Paleo- Asian Ocean in the CAOB (XMOB) and the onset of the subduction of the Pacific ocean, which are controversial in the international community. This paper presents new detrital zircon U-Pb dating and Hf isotopic data of the Early Mesozoic strata in NE North China Craton in order to address the above issues. The authors define that the Early Triassic deposition was

controlled primarily by southward subduction of the Paleo-Asian oceanic plate beneath the NCC, and collision between the NCC and the Yangtze Craton (YC). They also infer that rapid uplift of the XMOB and the onset of subduction of the Paleo-Pacific Plate beneath Eurasia occurred in the Early Jurassic. I enjoy reading this manuscript featured with good quality of zircon U-Pb ages and Hf isotopic data. Therefore, I would like to recommend the manuscript to bef published in Solid Earth Discussions after a minor to moderate revision.

Specific comments * While the closure of the Paleo-Asian ocean has been nicely defined by the data in this work, consistent with some other evidence in the literature, the reader is not easier to catch another important conclusion that the onset of subduction of the Paleo-Pacific Plate beneath Eurasia occurred in the Early Jurassic. Although it is mentioned that there is the presence of an Early Jurassic accretionary complex, but it is not on any of the figures, which makes it hard to follow. More importantly, why and how the data in the current work verify the Early Jurassic subduction is not clearly explained. Please have more arguments or explanations.

* XMOB is well known in China, but less commonly used in the international community, while the Central Asian Orogenic Belt (CAOB) is more commonly used. I would like to suggest, in order to enhance the international readship for the current work, at least add CAOB after it, namely, XMOB (eastern CAOB) or any other types of connection between these two terms.

P. 10, L 15: better use "confirm" instead of "determine".

P. 14, L 10: add some refs after "remain controversial".

Fig. 1: spell all the acronyms. Spell JJOB in caption. Also should mark the various terranes (such as JM: Jiamusi Massif; KM: Khanka Massif) in this map otherwise the tectonic evolution model with these terranes in Fig. 12 is hard to follow.

Fig. 5: what are the numbers in circles? Please explain either in captions or directly on

[Figure]

the figure(s).

Fig. 10: Please mark CAOB in the north and Qinling-Dabie in the south, together with all the faults in particular the Tan-Lu Fault, in the map. "Upper" and "lower" are spelled sometimes all small letters, but sometimes with first latter capital; make them in consistency.

Fig. 11: add the ages of these formations or a time bar beside them so that the reader can easily judge the detrital zircon ages and their provenance implications.

Fig. 12: spell all the acronyms directly in the figure or sub-figures so that the reader can immediately catch the tectonic scenario. All subduction zones should be marked in this figure or sub-figures. The presence of an Early Jurassic accretionary complex mentioned in the text should be marked. North mark is too distorted to see; just N is enough.

Wenjiao Xiao

---

## Author Comment (AC1) · 7 Oct 2018

**Response to the comments 1**

We thank the Editor-in-Chief's and reviewer 1's comprehensive and constructive reviews, it is very important to improve the quality of this paper. Based on the Editor-in-Chief's and reviewer 1's suggestions and comments, the manuscript has been revised.

This attachment includes: 1) Response to the comments 1; 2) The revised manuscript (document in revised mode). In the revised manuscript, the original Figure 1 (a, b) is split into new Figure 1 and Figure 2, the other figures are still numbered sequentially, so 13 figures are provided in this submission.

Please note this also includes track changes made after Reviewer 2.

**Anonymous Referee 1 (reviewer's comments in bold)**

**The timing for final closure of the Paleo-Asian Ocean and onset of subduction of the Paleo-Pacific Plate beneath Eurasia is a very important topic in eastern Asian geological researches, but is still poorly constrained. In this manuscript, the authors presented new detrital zircon U-Pb dating and Hf isotopic data of the Early Mesozoic strata in northeastern North China Craton, with an aim to constrain their deposition ages and provenances. The authors proposed that onset of subduction of the Paleo-Pacific Plate beneath Eurasia occurred during the Early Jurassic and the final closure of the Paleo-Asian Ocean likely occurred in the Middle Triassic. The zircon U-Pb ages and Hf isotopic data are in good quality. Therefore, the manuscript is worthy of publication in Solid Earth Discussions after a moderate revision. My comments and suggestions are listed below:**

**1. In several places (Page 1, Line 10; Page 2, Lines 24-25), the authors mentioned their using detrital zircon data "to reconstruct the early Mesozoic tectono-paleogeography of the region". This is a very difficult task. Although the detrital zircon U-Pb data presented in the manuscript is good in quality and valuable, there is no any field data from sedimentary analyses (such as paleocurrent direction, sedimentary facies). Therefore, I don't think that the authors can reconstruct the early Mesozoic tectono-paleogeography only by their limited detrital zircon data from the early Mesozoic strata.**

Response:

We agree with the reviewer's suggestion. Indeed, sedimentary analyses are absent in our manuscript, which is always important to reconstruct tectono-paleogeography. However, more and more researches have indicated that detrital zircon geochronology has become a powerful tool for provenance analysis, particularly for helping to constrain paleogeography, tectonic reconstructions, and crustal evolution (Ross and Bowring, 1990; Gehrels and Dickinson 1995; Gehrels et al., 1995, 2002; Cawood and Nemchin, 2001). Therefore, we consider that detrital zircon geochronology also is critical in paleogeographic and tectonic reconstruction, and provides valuable information for reconstruction of tectono-paleogeography.

Changes:

(1) We emphasize the meaning of detrital zircon geochronology in reconstruction of paleogeography, and add four references (Page 2, Lines 14-17).

(2) According reviewer's suggestion, to be more rigorous, we use "understand" instead of "reconstruct" in the abstract (Page 1, Line 10), and delete "Reconstruction" in the headline of section 5.4 (Page 14, Line 24), respectively.

**2. In the Abstract (Page 1, Lines 11-13, 16-17, 19-20), Conclusions (Page 16, 7-10) and sections "5.3 Provenance of early Mesozoic strata in the northeastern NCC" and "5.4 Reconstruction of the early Mesozoic tectono-paleogeography of the northeastern NCC", the authors tried to use the percentages of detrital zircon U-Pb ages to give a quantitative analysis of sedimentary provenances. However, since zircon is only a very minor mineral in the sedimentary rocks, 42% detrital zircons with characteristics of the northern margin of the NCC do not mean that the 42% sediments were sourced from the northern margin of the NCC. Therefore, I don't think you can give a quantitative constraints on the sedimentary provenances of early Mesozoic sedimentary rocks.**

Response:

We agree with the reviewer's suggestion. Indeed, it is hard to give a quantitative analysis in the research of sedimentary provenances only based on the detrital zircon geochronology. In our manuscript, the data only from six samples just give a qualitative analysis of sedimentary provenance rather than a specific quantitative analysis.

Changes:

We carefully consider various expressions to indicate the qualitative result in provenance, compared with abstract words, the expression of precise percentage may be easier to understand, which will enable readers to catch the important information directly. However, in order to avoid misunderstanding the meaning of data, we explain specially that these data can't represent a quantitative result in the section 5.3 (Page 13, Lines 12, 13).

**3. Page 2, Lines 16-17: Words "poorly constrained" are better than "relatively unconstrained".**

Response:

We agree with the reviewer's suggestion.

Changes:

Revised (Page 2, Lines 21, 22).

**4. Page 3, Line 17: Neoproterozoic mafic magmatism around 900 Ma have been identified from the NCC in recent years by many studies (Liu et al., 2006, Chin. Sci. Bull. 51, 2375–2382; Gao et al., 2010, Geol. Bull. China 29, 1113–1122; Peng et al., (011, Lithos 127, 210–221; 2011, Gondwana Res. 20, 243–254; Wang et al., 2012, Sci. China Earth Sci. 55, 1461–1479; Zhang et al., 2016, Precambrian Research 272, 203–225).**

Response:

We carefully refer these literatures. Indeed, Neoproterozoic mafic magmatism around 900 occur in the NCC, but less than 800 Ma magmatism is absent in the NCC.

Changes:

"Neoproterozoic mafic magmatism" has been instead of "late Neoproterozoic" (Page 3, Line 27).

**5. Page 5, Lines 5-16: Is this the volume percentage of minerals? If then, please change "%" to "vol. %".**

Response:

Yes, this's the volume percentage of minerals.

Changes:

We change "%" to "vol. %" (Page 5, Lines 17-33; Page 6, Lines 1-5).

**6. Page 16-25: In the reference list, the authors list almost 22 papers from their group. I suggest the authors add some relevant references from other groups.**

Response:

We agree with the reviewer's suggestion.

Changes:

We refer literatures again carefully, and add 22 literatures from at least 4 different groups in this field. They are added in the text (Page 1, Lines 27, 28; Page 2, Lines 11, 12, 16, 17; Page 3, Line 8; Page 4, Line 6; Page 14, Lines 30, 31;Page 16, Line 22), and the reference list (Pages 18-28).

**7. Figure 10: Check the strata column of Xiabancheng Basin. Xingshikou and Nandaling formations are common in the Xiabancheng Basin, but is not shown in the strata column.**

Response:

We refer Regional Geology of Hebei Province (Geological Publishing House, 1989), and check the early Mesozoic strata in the Xiabancheng area. As reviewer's opinion, the Xingshikou and Nandaling formations are widespread in the Xiabancheng Basin. Recently, Meng et al. (2018, under review) give the ages of volcanics in the Xingshikou Fm (225 ± 1 Ma) and Nandaling Fm (227.6 ± 2 Ma), indicating that they belong to the Late Triassic rather than the Early Jurassic. In order to avoid confusing them with the Xingshikou and Nandaling formations established originaly, Meng et al. (2018) renamed these formations as the Xiaolanwuo Fm and Nandaling Fm in the Xiabancheng Basin, respectively.

Changes:

We add the parentheses under the Xiaolangwuo Fm and Nandaling Fm in the stratigraphic column of the Xiabancheng Basin, repectively, to explain this relationship (Fig. 11).

[revised manuscript text omitted]

(a) Early Triassic

(b) Late Triassic

(c) Early Jurassic

---

## Author Comment (AC2) · 7 Oct 2018

**Response to the comments 2**

We thank the Editor-in-Chief's and reviewer 2's comprehensive and constructive reviews, it is very important to improve the quality of this paper. Based on the Editor-in-Chief's and reviewer 2's suggestions and comments, the manuscript has been revised.

This attachment includes: 1) Response to the comments 2; 2) The revised manuscript (document in revised mode). In the revised manuscript, the original Figure 1 (a, b) is split into new Figure 1 and Figure 2, the other figures are still numbered sequentially, so 13 figures are provided in this submission.

Please note this also includes track changes made after Reviewer 1.

<table>
<tr><td>Referee 2 (reviewer's comments in bold)</td></tr>
</table>

**General comments: The early Mesozoic evolution of multiple tectonic regimes in the northeastern North China Craton is of key importance for a better understanding of the tectonics of E Asia including the final closure of the Paleo- Asian Ocean in the CAOB (XMOB) and the onset of the subduction of the Pacific ocean, which are controversial in the international community. This paper presents new detrital zircon U-Pb dating and Hf isotopic data of the Early Mesozoic strata in NE North China Craton in order to address the above issues. The authors define that the Early Triassic deposition was controlled primarily by southward subduction of the Paleo-Asian oceanic plate beneath the NCC, and collision between the NCC and the Yangtze Craton (YC). They also infer that rapid uplift of the XMOB and the onset of subduction of the Paleo-Pacific Plate beneath Eurasia occurred in the Early Jurassic. I enjoy reading this manuscript featured with good quality of zircon U-Pb ages and Hf isotopic data. Therefore, I would like to recommend the manuscript to bef published in Solid Earth Discussions after a minor to.**

**1. While the closure of the Paleo-Asian ocean has been nicely defined by the data in this work, consistent with some other evidence in the literature, the reader is not easier to catch another important conclusion that the onset of subduction of the Paleo-Pacific Plate beneath Eurasia occurred in the Early Jurassic. Although it is mentioned that there is the presence of an Early Jurassic accretionary complex, but it is not on any of the figures, which makes it hard to follow. More importantly, why and how the data in the current work verify the Early Jurassic subduction is not clearly explained. Please have more arguments or explanations.**

Response:

We agree with the reviewer's suggestion. The onset of subduction of the Paleo-Pacific Plate beneath Eurasia is not expressed clearly in the section 5.4.3 and conclusion. Although we provide the data in good quality in this work, more evidences should be detailedly provided to constrain the tectonic evolution. Therefore, we rewrite section 5.4.3 and redraw Figure 1 in the revised manuscript.

Changes:

(1) The sudden change in provenance is detailedly explained, and its initial mechanism is inferred step by step (Page 16, Lines 5-11).

(2) The Early Jurassic accretionary complex in China, Japan and Russian Far East and Early Jurassic igneous rocks in eastern NE China are shown in the new Figure 1, and the evidences related to the onset of the subdcution of the Paleo-Pacific Plate have been provided (Page 16, Lines 5-24; Page 16, Lines 1-5).

(3) The onset of subduciton of the Paleo-Pacific Plate begin with the Early Jurassic is explained in the conclusion (3) (Page 17, Lines 17-19).

(4) The original Figure 1 (a, b) has been split into new Figure 1 and Figure 2 in the revised manuscript, and new Figure 1 shows Geological sketch map of the northeastern North China Craton. A more detailed tectonic units is shown, such as Erguna, Xing'an, Songnen-Zhangguangcai Range, Jiamusi, and Khanka massifs, these massifs are also marked in the new Figure 13. The corresponding description in words about new Figure 1 is also added in the revised manuscript (Page 3, Lines 5-8).

**2. XMOB is well known in China, but less commonly used in the international community, while the Central Asian Orogenic Belt (CAOB) is more commonly used. I would like to suggest, in order to enhance the international readship for the current work, at least add CAOB after it, namely, XMOB (eastern CAOB) or any other types of connection between these two terms.**

Response:

We agree with the reviewer's suggestion.

Changes:

It's done (Page 3, Lines 4-6).

**2. P.10, L 15: better use "confirm" instead of "determine".**

Response:

We agree with the reviewer's suggestion.

Changes:

It's done (Page 10, Line 30).

**3. P. 14, L 10: add some refs after "remain controversial".**

Response:

We agree with the reviewer's suggestion.

Changes:

We add 11 references after "remain controversial" (Page 14, Lines 31, 32).

**4. Fig. 1: spell all the acronyms. Spell JJOB in caption. Also should mark the various terranes (such as JM: Jiamusi Massif; KM: Khanka Massif) in this map otherwise the**

**tectonic evolution model with these terranes in Fig. 12 is hard to follow.**

Response:

    We agree with the reviewer's suggestion.

Changes:

    (1) We spell JJOB in caption of the new Figure 1.

    (2) The original Figure 1 (a, b) is split into new Figure 1 and Figure 2 in the revised manuscript, and new Figure 1 shows Geological sketch map of the northeastern North China Craton. A more detailed tectonic units is shown, such as Erguna, Xing'an, Songnen-Zhangguangcai Range, Jiamusi, and Khanka massifs, these massifs are also marked in the new Figure 13.

**5. Fig. 5: what are the numbers in circles? Please explain either in captions or directly on the figure(s).**

Response:

    The numbers in circles show the sequence of bed in the section.

Changes:

    We explain them in captions of the new Figures 3, 4, 5.

**6. Fig. 10: Please mark CAOB in the north and Qinling-Dabie in the south, together with all the faults in particular the Tan-Lu Fault, in the map. "Upper" and "lower" are spelled sometimes all small letters, but sometimes with first latter capital; make them in consistency.**

Response:

    We agree with the reviewer's suggestion.

Changes:

    We mark CAOB in the north and Qinling-Dabei in the south, together with Tan-Lu faults in the map (Yitong-Yilan, Dunhua-Mishan, Linjinjiang), and unify the initial of "Upper" and "lower" into lowercase in the new Figure 11.

**Fig. 11: add the ages of these formations or a time bar beside them so that the reader can easily judge the detrital zircon ages and their provenance implications.**

Response:

    We agree with the reviewer's suggestion.

Changes:

    It's done (Figure 12).

**Fig. 12: spell all the acronyms directly in the figure or sub-figures so that the reader**

**can immediately catch the tectonic scenario. All subduction zones should be marked in this figure or sub-figures. The presence of an Early Jurassic accretionary complex mentioned in the text should be marked. North mark is too distorted to see; just N is enough.**

Response:

We agree with the reviewer's suggestion.

Changes:

(1) According to the new figure 1, we also marked these microcontinental massifs in the Figure 13 to provide a more detailed tectonic scenario.

(2) All the acronyms are directly spelled except for the NCC.

(3) Suture zone and faults are marked.

(4) The Heilongjiang Complex is marked in the Figure (13c), and exposed along the Mudanjiang fault.

(5) The North mark is distorted for a special purpose to show the 3D effects, but it is hard to see. Therefore, we simplify the mark to easy to see, but continue to have the 3D effects to conform to vision of the figure.

[revised manuscript text omitted]

246±2 Ma 253±3 Ma (Hf isotope)
160° 15LJ4-6  15LJ4-11
252±1 Ma
16LJ6-1

0  100  200m

a  sandstone andesite  b sandstone  c conglomerate

silt shale   siltstone   sandstone   feldspathic quartz sandstone   gravel-bearing sandstone   conglomerate   andesite

T₁h Heisonggou Fm   Pt₂ Mesoproterozoic strata   ripple mark   plant fragment fossils   plant fossils   fault   photo location

**Figure 5**

[Figure]

**Figure 6**

[Figure]

**Figure 7**

[Figure]

**Figure 8**

[Figure]

**Figure 9**

[Figure]

**Figure 10**

[Figure]

**Figure 11**

[Figure]

**Figure 12**

[Figure]

**Figure 13**

[Figure]

(a) Early Triassic

(b) Late Triassic

(c) Early Jurassic

---

## Editor Decision (ED1)

[revised manuscript text omitted]

---

## Author Response (AR2)

Dear Prof. Federico Rossetti,

We thank your comprehensive and constructive reviews, it is very important to improve the quality of this paper. Based on your suggestions and comments, the manuscript and figures have been revised.

The responses to your comments are as follows (comments in bold):

**(1) The Introduction section needs to present a more exhaustive description of the scientific rationale and of the geological issue pursued in this study. In other words: why is it important to study these basin? Which the gap of knowledge? which information can they provide at regional scale? Which the expected advancement in the state of knowledge?**

Response: We agree with your comments. The introduction is very important to present description of the scientific rationale and of the geological issue.

Change: Based on your suggestions, we rewrite introduction section. 1) The significance of the northeastern NCC is added in the beginning of the introduction; 2) The debates of the previous studies are explained clearly, including their disadvantages; 3) The research status of early Mesozoic strata in the northeastern NCC is summarized, and the questions are proposed; 4) The implication for the evolution of the sedimentary basins is emphasized, and the aims are proposed.

**(2) A Material and Method section is missing. This section can help the reader to follow the scientific rationale of the study. This section should indicate the available material and the methods (including field work, stratigraphy logging...) adopted to achieve the main aims of the study. Maybe the analytical protocols should be moved to an Appendix, but I leave this to the Authors.**

Response: We agree with your comments. This section is necessary to the reader.

Change: 1) We add the Materials and Methods section after the Geological background (Page 5, Line 22); 2) A summary introduction is added in the beginning of this section (Page 5, Lines 23-32; Page 6, Lines 1-2); 3) The structure of this section is reorganized, the previous 'Sampling description' and 'Analytical methods' are classified into this section (Page 6, Line 3; Page 7, Line 4). The sampling description belongs to the detailed interpretation for materials.

**(3) Text needs some improvements (see attached commented pdf file).**

Response: We agree with your comments.

Change: Text has been revised based on your comments, and a few contents are annotated and/or explained (please see attached revised pdf file).

**(4) Figures needs some improvements (see the attached commented pdf file).**

Response: We agree with your comments.

Change: Figures have been revised based on your comments (please see attached revised pdf file).

Figure 1: We add the location of the Fusong-Changbai Bsain Grpoup and also refer to Fig. 1 in the text when mention it (Page 2, Line 22 ).

Figure 2: The color of the cross sections is replaced.

Figure 3: The word "diabase prophyrite" instead of "allgovite".

Figure 4: 1) The figure and text are enlarged; 2) The interpretation of the significance of the numbers is added in the figure caption.

Figure 5: 1) The word "diabase prophyrite" instead of "allgovite"; 2) The dotted line represents vegetation cover, which also is added in the figure captions; 3) The figures and text are enlarged.

Figure 6: 1) The figure and text are enlarged; 2) The interpretation of the significance of the numbers is added in the figure caption.

Figure 13: The titles of the three periods are renewed.

**Others:**

Table S1 has been revised, so it is submitted again. 1) The word "diabase prophyrite" instead of "allgovite" (Page 13, sample 15JFS2-1), and the data of Th/U is renewed (Pages 13-16, samples 16LJ8-1).

We hope that the present version can be put in publication process.

Sincerely,

Yini Wang, Wenliang Xu, Feng Wang, Xiaobo Li

[revised manuscript text omitted]

Figure 6

160°

227±1 Ma
16LJ1-1

184±2 Ma (Hf isotope) 182±1 Ma
15LJ1-2      16LJ3-1

Erdao Valley

① ② ③ ④ ⑤ ⑥ ⑦ ⑧ ⑨ ⑩ ⑪

0    100    200m

T₃ch    J₁y

a — andesite
b — Coal
c — feldspathic quartz sandstone
d — tuffaceous siltstone

shale        silt shale        siltstone        sandstone        feldspathic quartz sandstone        conglomerate        coal

tuffaceous siltstone        andesite        plant fossils        photo location        T₃ch Changbai Fm        J₁y Yihe Fm

[Figure]

**Figure 8**

[Figure]

[Figure]

[Figure]

**Figure 10**

[Figure]

**Figure 11**

[Figure]

**Figure 12**

[Figure]

**Figure 13**

(a) Early Triassic: southward subduction of the Paleo-Asian oceanic plate, and subduction and collision between the NCC and YC

(b) Late Triassic: final closure of the Paleo-Asian Ocean and post-collisional exhumation of the Su-Lu Orogenic Belt

(c) Early Jurassic: rapid uplift of the XMOB and the onset of subduction of the Paleo-Pacific Plate beneath Eurasia

---

## Author Response (AR3)

**Responses to Prof. Federico Rossetti**

Dear Prof. Federico Rossetti,

We thank your further reviews, it is very kindly of you to revise the English language. Based on your suggestions and comments, the manuscript have been revised.

The responses to your comments are as follows (comments in bold):

**(1) It mostly concerns the English language and manuscript organisation of the revised sections.**

We revise the English language and re-organise the revised sections. The titles of the 2.1 and 2.2 sections are deleted (Page 3, Line 19; Page 4, Line 32).

**(2) Please check for quality English editing before the re-submission.**

In order to check the quality English editing, **t**his revised manuscript has been edited again by a special English editing company (Stallard Scientific Editing) in New Zealand. They help to edit the English language and the format of the manuscript. After that, We check the whole manuscript again.

We look forward that the present version can be put in publication process.

Sincerely,

Yini Wang, Wenliang Xu, Feng Wang, Xiaobo Li

[revised manuscript text omitted]

---

## Author Response (AR4)

Responses to Prof. Federico Rossetti

Dear Prof. Federico Rossetti,

We thank your further reviews very much.

The responses to your comments are as follows (comments in bold):

(1) **Your manuscript can be accepted pending the technical corrections highlighted in the attached pdf file.**

Based on the technical corrections highlighted in the attached pdf file, we revised the manuscript.

**(2) Technical corrections are requested.**

Technical corrections are carried out, including references and figures. The references are checked again, and the figures are corrected in the format of longitude and latitude, as well as format of the unit symbol.

We look forward that the present version can be put in publication process.

Sincerely,

[revised manuscript text omitted]

(a) Early Triassic: southward subduction of the Paleo-Asian oceanic plate, and subduction and collision between the NCC and YC

(b) Late Triassic: final closure of the Paleo-Asian Ocean and post-collisional exhumation of the Su-Lu Orogenic Belt

(c) Early Jurassic: rapid uplift of the XMOB and the onset of subduction of the Paleo-Pacific Plate beneath Eurasia